# Breadcrumbs to the Goal: Goal-Conditioned Exploration from Human-in-the-Loop Feedback

**Marcel Torne**[1,2]    **Max Balsells**[3]    **Zihan Wang**[3]    **Samedh Desai**[3]
**Tao Chen**[1]    **Pulkit Agrawal**[1]    **Abhishek Gupta**[3]
[1]Massachusetts Institute of Technology    [2]Harvard University    [3]University of Washington
{marcelto,taochen,pulkitag}@mit.edu
{balsells,avinwang,samedh,abhgupta}@cs.washington.edu

## Abstract

Exploration and reward specification are fundamental and intertwined challenges for reinforcement learning. Solving sequential decision-making tasks requiring expansive exploration requires either careful design of reward functions or the use of novelty-seeking exploration bonuses. Human supervisors can provide effective guidance in the loop to direct the exploration process, but prior methods to leverage this guidance require constant synchronous high-quality human feedback, which is expensive and impractical to obtain. In this work, we present a technique called Human Guided Exploration (HuGE), which uses low-quality feedback from non-expert users that may be sporadic, asynchronous, and noisy. HuGE guides exploration for reinforcement learning not only in simulation but also in the real world, all without meticulous reward specification. The key concept involves bifurcating human feedback and policy learning: human feedback steers exploration, while self-supervised learning from the exploration data yields unbiased policies. This procedure can leverage noisy, asynchronous human feedback to learn policies with no hand-crafted reward design or exploration bonuses. HuGE is able to learn a variety of challenging multi-stage robotic navigation and manipulation tasks in simulation using crowdsourced feedback from non-expert users. Moreover, this paradigm can be scaled to learning directly on real-world robots, using occasional, asynchronous feedback from human supervisors. Project website at https://human-guided-exploration.github.io/HuGE/.

## 1 Introduction

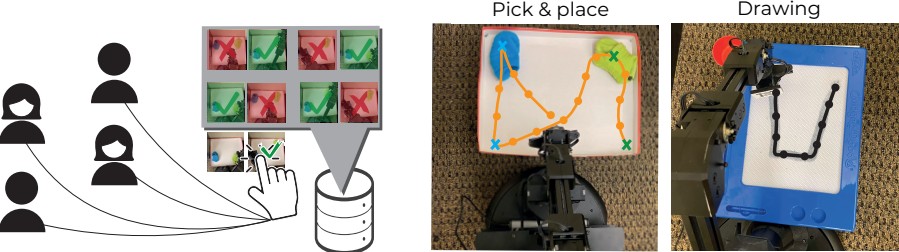

Figure 1: HuGE leverages noisy and asynchronous feedback from multiple non-expert humans to train robot control policies directly on the real world.

How should we teach agents a new task? A general method is to provide agents with a reward function and employ reinforcement learning. This may appear to be a straightforward plug-and-play

37th Conference on Neural Information Processing Systems (NeurIPS 2023).

procedure, yet in reality, it usually means iterative human intervention in designing reward functions to overcome exploration obstacles and circumvent potential reward hacking. The cyclical process of a human meticulously crafting a reward function, assessing the resulting behavior of the learned policy, and then refining the rewards to enhance performance can be a painstaking and inefficient process.

If an agent is able to effectively and autonomously explore its environment, it can learn new tasks using an easily designed sparse reward function [50, 20]. This would make the training process less reliant on humans designing detailed reward functions for "dense" supervision. Several strategies incentivize state coverage in exploration by expanding the frontier of visited states and explicitly rewarding discovering novel states.[46, 5, 21, 43]. When purely optimizing for state coverage, the agent *overexplores* parts of the state space that are not relevant to the task at hand [13]. This quickly becomes a futile effort when exploring in environments with large state spaces, suggesting the need for a more directed exploration strategy.

A promising avenue for directed exploration, without requiring a prohibitive amount of reward design, is in developing techniques that leverage non-expert human feedback to adjust the agent's training [55, 30, 11, 36]. This work is a step towards the overarching objective of learning new skills in the real-world through meaningful human-robot interactions. A challenge with techniques that optimize policies over learned reward functions is that humans may provide noisy feedback, or may not even know the optimal solution for the task at hand [18], rendering their feedback unreliable and biased, a problem which is exacerbated with a large number of non-expert supervisors. When this noisy and incorrect feedback is used to learn reward functions for RL, it can lead to learning biased and incorrect policies. Requiring human supervisors to provide high-quality feedback synchronously and frequently during the training process becomes prohibitively expensive. If we want to reduce the burden on supervisors it is important to ensure that inaccuracies, biases in the reward function are not reflected directly in the learned policy.

We note that reward functions in reinforcement learning serve two distinct purposes. One, determining which goal or behavior is optimal, and two, determining how to explore the environment to achieve this goal. In this work, we make the observation that these two independent purposes of reward functions can be decoupled. Human feedback can be used to guide exploration, while policy learning techniques like self-supervised reinforcement learning can learn unbiased optimal behaviors for reaching various goals from the data collected via exploration. This provides the best of both worlds: a way to guide exploration using non-expert human supervisors, while avoiding inheriting the bias from providing asynchronous, low-quality feedback.

Human Guided Exploration (HuGE) uses noisy, infrequent, and asynchronous binary human comparative feedback to guide exploration towards subsets of the state space closer to goal, while learning optimal policies via self-supervision. Rather than having humans directly label which states to visit, HuGE solicits users for binary comparisons between visited states to determine which are closer to the goal. These comparisons help train a goal selector which biases exploration towards chosen states that are more proximal to the objective. The process of exploration involves returning to these selected states and subsequently expanding the set of visited states from there. From this exploration data, we learn optimal policies for goal-reaching in a self-supervised manner using ideas from "hindsight" relabelling [24, 2, 28], hence avoiding reliance on any explicit reward function.

Iterating between exploration guided by the goal selector and self-supervised policy learning, HuGE expands the frontier of visited states until the desired target goals can be reached reliably. Opposed to directly using human feedback as a reward [16, 45], HuGE lets humans provide noisy, asynchronous, and infrequent feedback, as it is simply used to guide exploration, not directly as an objective for policy learning. In the worst case, when human feedback is incorrect or absent, the exploration is hampered but the policy learning scheme is still convergent to the optimal solution. This way, humans are able to essentially "drop breadcrumbs", incrementally guiding agents to reach distant goals, as shown in Fig 3. This helps us solve more complex tasks than if solely relying on self-supervision and is able to overcome the optimization bias that is often encountered when optimizing over a learned and noisy reward function [16]. Overall, this work presents the following contributions:

**Guided Exploration from Noisy and Minimal Human Feedback**: We propose an algorithm for separating (noisy) human feedback from policy learning by utilizing comparative, binary human feedback to guide exploration while learning a policy through self-supervision. This ensures that we can derive optimal goal-reaching policies even from imperfect human feedback.

**Capability to learn from crowdsourced non-expert human feedback**: We show HuGE can learn from noisy, asynchronous and infrequent human feedback. We validate this by collecting crowd-sourced pilot data from over a hundred arbitrarily selected, non-expert annotators who interact with the robotic learning through a remote web interface.

**Demonstration with real-world policy learning**: We show that HuGE can learn policies on a robotic learning system in the real world, demonstrating effective exploration via human feedback and compatibility with pretraining from non-expert trajectory demonstrations as seen in Figure 1.

## 2 Related Work

Our work builds on techniques for exploration methods, goal-conditioned, and human-in-the-loop RL, but presents a technique for learning from noisy, asynchronous and infrequent human feedback. We present the connections to each of these sub-fields below:

**Exploration in RL:** While exploration is a widely studied subfield in RL [10, 43, 5, 37, 9, 21], this is typically concerned with either balancing the exploration-exploitation tradeoff [4] or maximizing state coverage [5, 10, 37]. This contrasts with our goal of performing *targeted* exploration, informed by human feedback. Our work is related to ideas from Go-Explore [21], which explores by maintaining an archive of visited states, returning to states with a low-visitation count, and performing random exploration from there. However, this results in redundant overexploration shown in Figure 2 under the novelty-based exploration paradigm. In contrast, HuGE is able to leverage human feedback to only explore in promising directions as directed by human supervisors, significantly improving sample efficiency.

For goal-reaching problems, alternatives for exploration include self-supervision techniques such as hindsight relabeling [2, 24, 35]. By learning goal-reaching behavior in *hindsight* for reached states, these methods obtain dense supervision, albeit only for the states that were actually reached. This has been shown to empirically aid with exploration in certain cases, via policy generalization across goals. However, as shown in Figure 3, these methods suffer from *underexploration*, since the policy generalization across states can be challenging to control. In contrast, HuGE does not rely on arbitrary policy generalization, instead revisiting already *reached* states at the frontier of visited states and performing further exploration from these states.

**Goal-conditioned reinforcement learning:** Goal-conditioned RL algorithms [34, 24, 35, 2, 33, 28, 37] are multi-task RL methods where various tasks are defined as reaching different goal states. A number of different approaches have been proposed for goal-conditioned RL - learning with hindsight relabeling [2, 19, 33, 28], learning latent spaces for reward assignment [42, 48], learning dynamical distances [27, 22] and goal conditioned imitation learning [25, 35, 24, 40]. While these algorithms are able to solve tasks with a simple component of exploration, they can struggle with tasks with complex sequential nature, requiring complex machinery such as hierarchical architectures [33, 39]. In contrast, our work shows the ability to solve sequential goal-reaching problems with standard techniques, just using occasional comparative feedback from human supervisors to guide exploration.

Perhaps most closely related to our work is the paradigm introduced in DDL [27] where a human supervisor occasionally selects which states to explore from. This state is used to define the reward function, which when combined with self-supervised Q-learning, can aid with exploration for goal-conditioned RL problems. However, exploration is limited to exactly the state selected by the human, whereas in HuGE, learning the parametric goal selector allows the exploration frontier to continue expanding and decreases the overall amount of human feedback needed (Section 5, Figure 8).

**Reinforcement Learning from Human Feedback (RLHF):** RLHF typically characterizes a class of methods that learn reward models from binary human comparative feedback for use in RL. These techniques have seen use in better aligning language models, and guiding learning of embodied agents in simulation with expert feedback [16, 7, 8]. In much of this work, humans are assumed to provide high-quality and frequent feedback to learn rewards. In HuGE, we take a complementary view and show that by considering human feedback in a self-supervised policy learning setting, much of the burden can be placed on self-supervision, rather than on receiving high-quality human feedback. More broadly, human in the loop RL is a well explored concept, with various interfaces ranging from preferences [16, 7, 8, 31] to scalar rewards [29], language based corrections [53], binary right/wrong signals [12] and even sketching out rewards [11]. While we restrict the study

in this work to binary comparative feedback, exploring how the underlying principles in HuGE (of decoupling human feedback and self supervised policy learning) can be extended to other forms of feedback is an exciting avenue for future work.

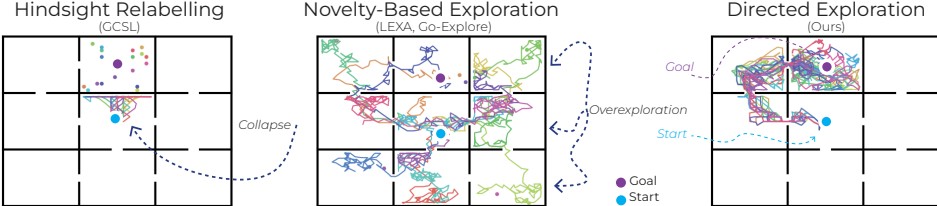

Figure 2: Comparison of exploration types. **hindsight relabelling**: suffers from exploration *collapse*, **novelty-based exploration** suffers from *overexploration*, we propose directed exploration from human feedback *Ours*.

## 3 Problem Setup and Preliminaries

This work solves goal reaching tasks by using goal-conditioned policy learning methods. The goal reaching problem is characterized by the tuple $\langle \mathcal{S}, \mathcal{A}, \mathcal{T}, \rho(s_0), T, p(g) \rangle$, adopting the standard MDP notation with $p(g)$ being the distribution over goal states $g \in \mathcal{S}$ that the agent is tasked with reaching. We assume sampling access to $p(g)$. We aim to find a stationary goal-conditioned policy $\pi(\cdot|s, g)$: $\mathcal{S} \times \mathcal{S} \to \Delta(\mathcal{A})$, where $\Delta(\mathcal{A})$ is the probability simplex over the action space. We will say that a goal is achieved if the agent has reached the goal at the end of the episode (or is within $\epsilon$ distance of the goal). The learning problem can be characterized as that of learning a goal conditioned policy that maximizes the likelihood of reaching the goal $\pi \leftarrow \arg\max_\pi J(\pi) = \mathbb{E}_{g \sim p(g)} \left[ P_{\pi_g} (s_T = g) \right]$.

**Goal-conditioned Policy Learning:** Goal-conditioned RL methods approach this problem using the reward function $r_g(s) = \mathbb{1}(s = g)$, defining a sparse reward problem: $\pi \leftarrow \arg\max_\pi \mathbb{E}_{\tau \sim \pi(\cdot|s,g), g \sim p(g)} \left[ \sum_{t=1}^{H} \gamma^t r_g(s_t) \right]$. This problem can be difficult to solve with typical RL algorithms [26, 52, 23] because the training process is largely devoid of learning signals.

### 3.1 Why is exploration in goal-conditioned reinforcement learning challenging?

**Hindsight relabelling**: To circumvent the challenge of sparse rewards in goal-conditioned reinforcement learning, the structure of goal-reaching problems can be exploited using the hindsight relabeling technique [28, 1, 3]. Hindsight relabeling leverages the insight that transitions that may be suboptimal for reaching "commanded" goals $g$, may be optimal in *hindsight* had the actually reached states $s_T$ been chosen as goals. This allows us to relabel a transition tuple $(s, a, s', g, r_g(s))$, with a hindsight tuple $(s, a, s', g', r_{g'}(s))$, where $g'$ can be arbitrary goals chosen in hindsight. When $g'$ is chosen to be states $s$ actually visited along a trajectory, the reward function $r_s(s) = 1$ provides a dense reward signal for reaching different goals $g = s$, which can be used to supervise an off-policy RL algorithm [3] or a supervised learning algorithm [1, 40, 47, 24].

The key to exploration in these methods is *generalization* - even though the desired goal distribution $p(g)$ is not accomplished, learning policies on the *accomplished* goal distribution $p(g')$ influences exploration towards $p(g)$ via generalization between $g'$ and $g$. However, this form of exploration is unreliable across domains and goal representations. As shown in Figure 2 and in Appendix E, this often *underexplores* the environment without accomplishing the target goal.

**Novelty-based exploration**: On the other hand, pure novelty-based exploration [21, 51, 5, 44] is also ineffective, since it performs task agnostic exploration of the entire state-space, thereby *overexploring* the environment (Fig 2). This becomes intractable with large state and action spaces.

**Directed exploration (Ours)**: A practical solution would explicitly encourage exploration but directed towards the goal. In the following sections, we will argue that providing noisy, asynchronous and infrequent binary comparison feedback by a variety of human users can provide this directed exploration, without inheriting the noise and bias of the provided feedback.

## 4 HuGE: Guiding Exploration in Goal-Conditioned RL with Human Feedback

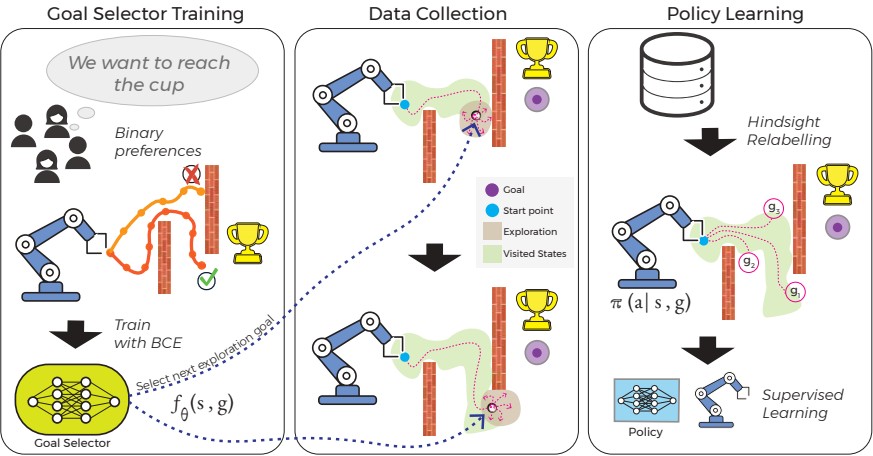

Figure 3: Overview of HuGE. We train a goal selector, from human feedback through state comparisons, to perform directed exploration in self-supervised learning.

The key idea behind HuGE is to leverage (noisy) human feedback for guiding exploration and data collection, but to decouple this from the process of learning goal-reaching policies from the collected data. We show how this can be done in a self-supervised way, allowing unbiased goal-reaching policies to be learned, while human feedback can still be useful in guiding exploration. We describe each component of our proposed algorithm below.

### 4.1 Decoupling Human Feedback from Policy Learning for Goal Reaching Problems

While human feedback has been shown to be an effective tool to guide exploration [16, 8], policies obtained by maximizing reward functions learned from noisy and infrequent human feedback can be very suboptimal, often getting trapped in local optima. This places a significant burden on human supervision, making it challenging for non-expert, casual supervisors to guide robot learning. *How can we enable policy learning from infrequent and noisy human feedback?*

Our key insight is that human feedback and policy learning can be disentangled by decoupling the process of exploration from the policy learning, while only leveraging the human feedback on the exploration process. In this new paradigm, where the generation of exploration data does not necessitate optimality, we can harness guiding signals originating from noisy human feedback, yet still effectively learning optimal goal-conditioned policies.

In particular, for human guided exploration we build on the idea of frontier expansion [21, 37, 14], where exploration is performed by revisiting states at the frontier of visited states and then continuing exploration from these. This technique would maintain a frontier $\mathcal{F}$ that consists of the set of all visited states. We can then select a previously visited state for the policy to revisit, which we will call *breadcrumb* state $g_b$, by querying a "goal selector" on $\mathcal{F}$, that is learned from (noisy) human feedback and defined in detail in 4.2. In this way, human feedback can be used to "softly" guide exploration towards promising states $g_b$. Frontier *expansion* can then performed by executing random exploration from the reached breadcrumb state $g_b$, and adding the collected data to the replay buffer for self-supervised policy learning.

Given data collected by frontier expansion directed using human feedback, goal-reaching policies can then be learned by leveraging self-supervised policy learning techniques [24, 2, 35] that are able to learn policies from collected data using "hindsight" relabeling and supervised learning. Importantly, this self-supervised policy learning procedure does *not* depend on the human feedback at all and is purely self-supervised. The decoupling of exploration and policy learning implies that even if this exploration is biased and imperfect, the self-supervised policy learns *unbiased* paths to reach all goals that were visited. In this way, cheap low-quality supervision from non-expert human feedback can

be used to ensure guided frontier expansion towards relevant subsets of the state space, while still being able to learn unbiased goal-reaching policies. Below, we delve into each of these components - guided frontier expansion and self-supervised policy learning and describe the overall algorithm in Algorithm 1.

## 4.2 Guiding Exploration from Human Comparative Feedback by Learning Goal Selectors

To direct exploration more effectively, we hypothesize that receiving small amounts of occasional guidance from non-expert human users can be beneficial, even if the feedback is biased and noisy. To leverage this feedback, we learn a parametric goal selection function directly from binary human comparison feedback. Learning a parametric goal selection function allows this feedback to be useful even when humans are providing feedback infrequently and asynchronously.

**Learning State-Goal Distances from Binary Comparisons**: We propose a simple interface between the human and the algorithm that simply relies on the human supervisor to provide binary comparisons of which of two states $s_1$, $s_2$ is closer (given a human's judgment) to a particular goal $g$ (as demonstrated in Fig 3 and Appendix Fig B.4). These binary comparisons can be used to train an unnormalized estimate of distances $f_\theta(s, g)$ by leveraging the Bradley-Terry model of choice [16, 8, 7]:

$$
\max_\theta \sum \mathbb{1}(s_1 > s_2 | g) \log \frac{\exp f_\theta(s_1, g)}{\exp f_\theta(s_1, g) + \exp f_\theta(s_2, g)} +
$$
$$
(1 - \mathbb{1}(s_1 > s_2 | g)) \log \frac{\exp f_\theta(s_2, g)}{\exp f_\theta(s_1, g) + \exp f_\theta(s_2, g)}
\tag{1}
$$

This objective encourages states $s$ closer to particular goals $g$ to have smaller $f_\theta(s, g)$. While this estimate may be imperfect and noisy, it can serve to bias exploration in promising directions.

**Using the Learned State-Goal Distances for Biasing Exploration:** To guide exploration, we can select which breadcrumb state $g_b$ to command and start exploring from during frontier expansion, by sampling states inversely proportional to their distance to the goal measured as, $\exp(-f_\theta(s, g))$, where $f$ is learned above. This encourages guiding exploration towards states that have a lower estimated distance to the goal since these are more promising directions to explore, as indicated by the human comparisons. $g_b \sim p(g_b|g); p(g_b|g) = \frac{\exp{-\alpha f_\theta(g_b, g)}}{\sum_{g' \in \mathcal{D}} \exp{-\alpha f_\theta(g', g)}}$ (2) where $\mathcal{D}$ represents the set of reached states. Sampling through a *soft* distribution instead of choosing the state with the minimum estimated distance, can overcome noise and errors in the learned distance function, albeit at the cost of slower (yet convergent) exploration. The choice of softmax temperature $\alpha$ determines how much the exploration algorithm trusts the goal selector estimate (large $\alpha$), versus resorting back to uniform frontier expansion (small $\alpha$). The key to effectively using this state-goal distance function, is getting away from directly optimizing the reward function learned from human preferences, avoiding the aforementioned problems regarding the optimization of the reward function learned from human preferences and instead relying on self-supervision for policy training, Section 4.3.

## 4.3 Self-Supervised Policy Learning: Hindsight Relabeled Learning for Goal-Conditioned Policies

Given exploratory data collected by guided frontier expansion (Algorithm 2, Section 4.1), we can leverage a simple self-supervised learning scheme building on [1, 47, 24] for goal-conditioned policy learning. Given trajectories $\tau = \{s_0, a_0, s_1, s_2, \ldots, s_T, a_T\}_{i=1}^N$ we to construct a dataset of optimal tuples using: $\mathcal{D}_\tau = \{(s_t, a_t, g = s_{t+h}, h) : t, h > 0, t + h \leq T\}$(3). This relabeled optimal dataset can then be used for supervised learning: $J_{\text{Policy}}(\pi) = \mathbb{E}_{\tau \sim \mathbb{E}_g[\pi_{\text{old}}(\cdot|g)]} \left[ \sum_{t=0}^T \log \pi(a_t|s_t, g) \right]$ (4). This process can be repeated, iterating between collecting data, relabeling it, and performing supervised learning. As policy learning continues improving, the learned policy can be deployed to solve an expanding set of goals, eventually encompassing the desired goal distribution. The overall pseudocode of HuGE is shown in Algorithm 1. We note that this relabeled supervised learning scheme has been studied in past works [1, 47, 24], the unique contribution here being how low-quality human comparative feedback is used to guide exploration for self-supervised policy learning. While prior techniques such as [1, 47, 24] often struggle with exploration, the occasional human feedback can lead to significantly more directed exploration behavior for HuGE as compared to prior work.

**Boostrapping Learning from Trajectory Demonstrations:** While the system thus far has described a strategy that is applicable for learning policies from scratch, training HuGE from scratch can be prohibitively slow in the real world, and instead we may look to finetune already pre-trained "foundation" models or existing goal-directed policies using HuGE, or to pretrain from human demonstrations as considered in a huge plethora of prior work [49, 32, 6, 41].

Given that the self-supervised policy learning method in HuGE is fundamentally based on iterated supervised learning, a natural way to ensure the practicality of the method is by simply initializing the policy $\pi$ using supervised imitation learning on trajectory demonstrations (or from other sources if available). While this technique is effective on policy pretraining, it fails to account for the goal selection model $f_\theta(s, g)$. A simple modification to the training process allows *also* for effective pre-training the state-goal distances model $f_\theta(s, g)$ from demonstrations. Concretely, since demonstrations are typically monotonically decreasing in terms of effective distance to the goal, a single trajectory $\tau$ can be reinterpreted as a set of $\frac{n(n+1)}{2}$ comparisons, where later states in the trajectory are preferred to earlier ones: $\{s_{t_1} < s_{t_2} : t_1 < t_2 \forall s_{t_1}, s_{t_2} \in \tau\}$. These comparisons can then be used to pre-train the goal-selector, as discussed in Equation 1. Our proposed technique is not a replacement for learning from demonstrations, but rather serves a complementary role. As we show experimentally, this combination of HuGE and pre-training from non-expert demonstrations actually enables learning in the real world under practical time constraints.

---

**Algorithm 1** HuGE: Guided Exploration with Human Feedback

1: **Input:** Human $\mathcal{H}$, goal distribution $p(g)$
2: Initialize policy $\pi$, goal selector $f_\theta$, data buffer $\mathcal{D}$, goal selector buffer $\mathcal{G}$
3: **while** True **do**
4:     Sample goal $g \sim p(g)$
5:     $\mathcal{D}_\tau \leftarrow$ PolicyExploration$(\pi, \mathcal{G}, f_\theta, g, \mathcal{D})$
6:     $\mathcal{D} \leftarrow \mathcal{D} \cup$ RelabelTrajectory$(\mathcal{D}_\tau)$ (3)
7:     $\pi \leftarrow$ TrainPolicy$(\pi, \mathcal{D})$ (4)
8:     $\mathcal{G} \leftarrow \mathcal{G} \cup$ CollectFeedback$(\mathcal{D}, \mathcal{H})$ ( 4.2)
9:     $f_\theta \leftarrow$ TrainGoalSelector$(f_\theta, \mathcal{G})$ (1)

**Algorithm 2** PolicyExploration

1: **Input:** policy $\pi$, goal selector $f_\theta$, goal $g$, data buffer $\mathcal{D}$
2: $g_b \sim$ SampleBreadcrumb$(f_\theta, g, \mathcal{D})$(2)
3: $\mathcal{D}_\tau \leftarrow \{\}$
4: **for** $i = 1, 2, \ldots, N$ **do**
5:     $s \leftarrow s_0$
6:     **while** NOT stopped (see G.1) **do**
7:         Take action $a \sim \pi(a|s, g_b)$
8:     Execute $\pi_{\text{random}}$ for $H$ timesteps
9:     Add $\tau$ to $\mathcal{D}_\tau$ without redundant states
10: **return** $\mathcal{D}_\tau$

---

## 5 Experimental Evaluation

In this work, we show that HuGE learns to successfully accomplish long-horizon tasks, and tasks with large combinatorial exploration spaces through little human supervision. To demonstrate these experimentally, we test on several goal-reaching domains in simulation, shown in 4, in the MuJoCo [54] and PyBullet [17] simulators where we compare against state-of-the-art baselines. Furthermore, we show the benefits of our method by learning policies directly on a real-world LoCoBot robot. We additionally show the ability for HuGE to be used with an uncurated, large-scale crowdsourcing setup with non-expert human supervisors. With these experiments, we show **1)** HuGE outperforms prior work on solving long horizon goal-reaching tasks in simulation; **2)** HuGE is suitable for collecting crowdsourced, noisy and asynchronous feedback from humans all over the world with different backgrounds and education levels; **3)** HuGE is suited to learning in the real world with a robotic platform in practical time-scales, directly from visual inputs.

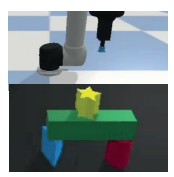 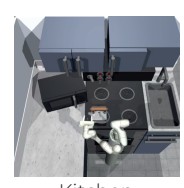 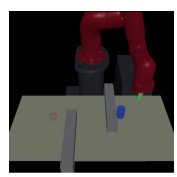 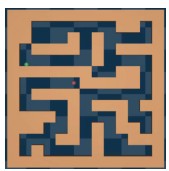 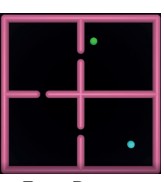

    Bandu      Block stacking     Kitchen     Pusher two walls    Maze    Four Rooms

Figure 4: Six simulation benchmarks where we test HuGE and compare against baselines. **Bandu**, **Block Stacking**, **Kitchen**, and **Pusher**, are long-horizon manipulation tasks; **Four rooms** and **Maze** are 2D navigation tasks, see Appendix D

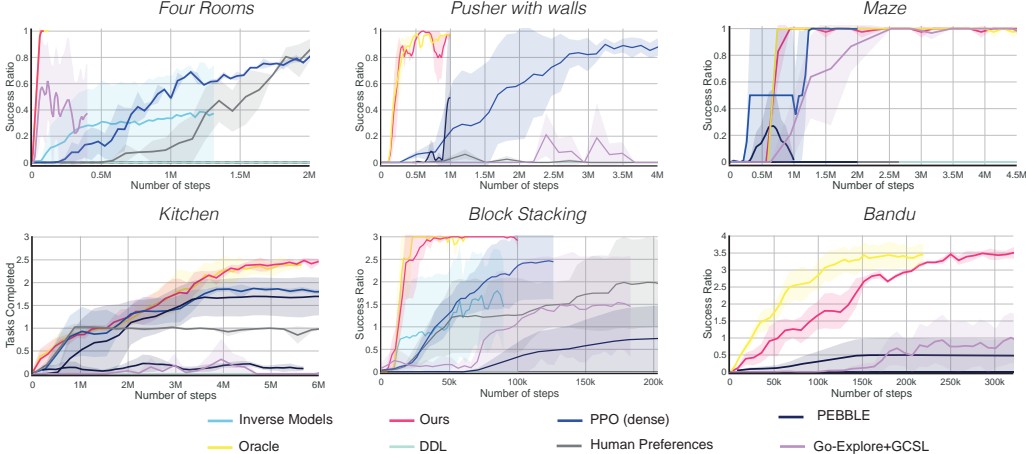

Figure 5: Success curves of HuGE on the proposed benchmarks compared to the baselines. HuGE outperforms the rest of the baselines, some of which cannot solve the environment while converging to the oracle accuracy. For those curves that are not visible, it means they never succeeded and hence are all at 0 (see E.10 for distance curves). The curves are the average of 4 runs, and the shaded region corresponds to the standard deviation.

## 5.1 Learning Goal-Conditioned Policies with Synthetic Human-in-the-Loop Feedback in Simulation

We consider goal-reaching problems in the six domains shown in Fig 4. These are domains with non-trivial exploration challenges —the agents must assemble a structure in a specific order without breaking it, navigate around walls, etc., and purely random exploration is unlikely to succeed. We evaluate HuGE compared to the baselines described in Appendix E on these domains. These baselines were chosen to compare HuGE with methods that perform different types of exploration, hindsight relabeling, and methods that use human preferences to learn a reward function instead, to highlight the benefits of decoupling exploration from human preferences and policy learning. We report the number of final goals reached successfully in Fig 5 as learning progresses. Only in these and the analysis experiments, the human feedback is synthetic (see Appendix D).

In Figure 5 and in Appendix E with further detail, we show HuGE learning a goal selector model to perform goal selection (*Ours*) matches, at convergence, the performance of using an *Oracle* always choosing the best goal to explore. Guiding exploration (*Ours*) is *significantly* better than techniques that perform indiscriminate exploration (*Go-Explore+GCSL*) [21, 37]. *Ours* also beats methods that purely rely on policy generalization and stochasticity to perform exploration and do not explicitly expand the frontier (*Inverse Models*) [1, 47, 24] which fail in complex exploration domains. Goal-conditioned reinforcement learning methods with *PPO* [52] and *sparse* rewards do not experience enough reward signals to actually learn directed behavior. In *PPO* with *dense* reward, we use the same reward as for generating the synthetic human labels and we observe, that in most cases the performance is lower. The reason is that we did not do any reward engineering for either PPO or HuGE, showing that *Ours* is more robust to simpler underlying reward functions than *PPO*, we analyze these results further in Appendix E.2. Similarly, using human feedback to bias exploration *Ours* beats using it to learn a reward function (*Human Preferences*)[16]. Finally, in Appendix C we show that pretraining HuGE with 5 noisy trajectories (*BC + Ours*) gives a significant decrease in the number of timesteps needed to succeed and beats plain Behavior Cloning (*BC*) and also compared to other baselines with demonstrations.

## 5.2 Learning Goal-Conditioned Policies with large-scale crowdsourced data collection

In this section, we show HuGE works on the kitchen environment from a crowdsourced data collection of 109 non-experts annotators labeling asynchronously and from all over the world, as shown in Figure 6. We spanned across three continents, having annotators living in 13 different countries, with ages ranging from 18 to 65+ and a variety of academic backgrounds, and we collected 1600 labels, we refer the reader to B.1 and B.2 for more details about the demographics. Each annotator

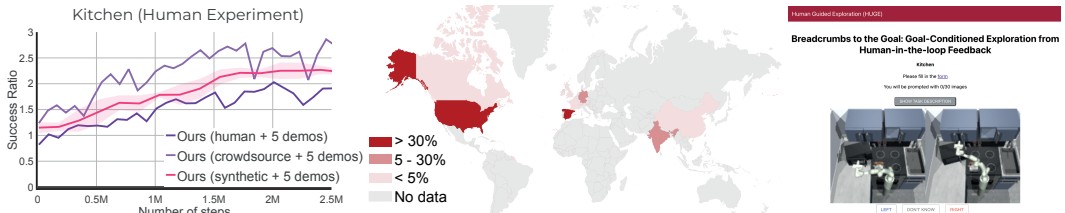

Figure 6: **left**: Crowdsourcing experiment learning curves for the kitchen, **middle**: human annotators spanned 3 continents and 13 countries, **right**: screenshot of the interface for data collection.

could provide labels at any time during the day, our recommendation was to provide 30 labels, which took on average less than 2 minutes of their time. We collected a total of 2678 labels for this crowdsourcing experiment beating *Ours* when labels are synthetically generated, This indicates that the crowdsourced data might provide more information than the simplistic reward function, designed to provide the synthetic feedback. We repeated the experiment collecting just 1670 labels from 4 annotators and saw that the crowdsourcing experiment also yield better results, suggesting that a wider variety of feedback does not damage performance. We refer the reader to Appendix B for additional results of HuGE from real human feedback on other simulated benchmarks, details on the platform we developed for these experiments, and where we clarify that this study was approved by the Institutional Review Board.

## 5.3 Learning Goal-Conditioned Policies in the real world

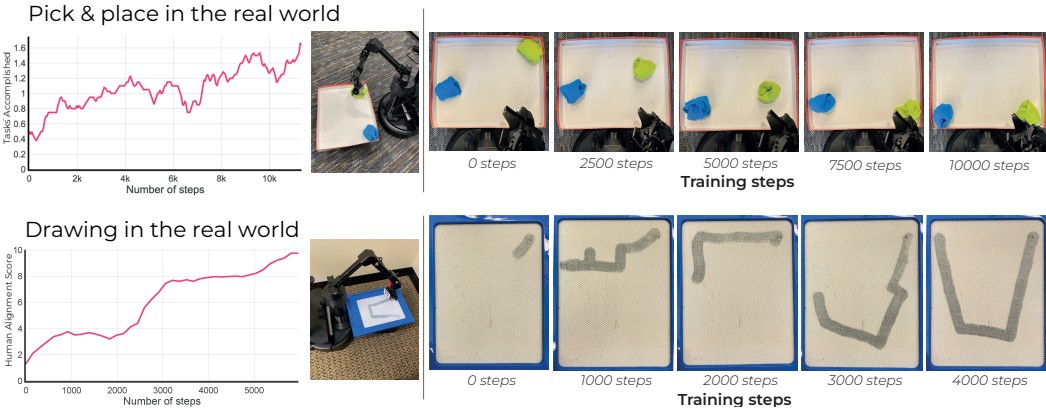

Figure 7: Accomplished goals at the end of 5 different evaluation episodes throughout training in the real world.

HuGE's qualities of being robust to **noisy** feedback and requiring **minimal** and **asynchronous** human supervision together with its self-supervised policy learning nature and the capability to be pretrained from trajectory demonstrations makes it suitable for learning in the real world. As shown in Fig 7, HuGE can learn policies for pick and place and drawing in the real world with a LoCoBot [38]. Some adaptations were made to run HuGE on real hardware such as changing the state space to images, instead of point space. In Appendix A we provide more results on HuGE from image space in the real world and simulation. We also pretrained our policy with 5 trajectory demonstrations (Appendix C. For both experiments, the robots learned directly on the real hardware. In the pick and place experiment, we collected around 130 labels across 20 hours of real-world robot training, whereas in the drawing one, we collected 150 labels across 6 hours.

## 5.4 Ablation Analysis

Lastly, we conducted a number of quantitative analysis experiments to better appreciate and convey the benefits of HuGE. In Figure 8, we show that learning a parametric goal selector (*Ours*) is more sample efficient than directly using the human-selected goals (*DDL* [27]). More concretely, learning a parametric goal selector needs 50% less human annotations than DDL [27]. The underlying

idea is that by deriving a goal selector from human feedback, we extract recurrent patterns and generalizations that can be redeployed in future rollouts. Conversely, without learning from this goal selector, we would overlook these patterns, leading to more frequent queries for human input.

One of the properties of HuGE is that it works from noisy human feedback. In Figure 8, we observe that despite adding large amounts of noise on the human annotations, HuGE still converges to the optimal solution. This supports the idea that when the human feedback is noisier, exploration will be less efficient and will take longer to converge nevertheless, still converging to an optimal policy.

Furthermore, in Figure 8 (*right*), we provide a visualization explaining why optimizing over the reward function (*Human Preferences* [16]) can provide non-optimal solutions. We observe reward functions learned from noisy human feedback are suboptimal, with multiple local maxima, and hence optimizing over them will potentially lead to the policy getting stuck in local optima. On the other hand, HuGE, which samples goals from this noisy learned reward function will still converge to the optimal solution. The explanation is that the goal selector will steer exploration towards the three modes uniformly, making exploration suboptimal but still converging to an optimal goal-conditioned policy reaching each one of the modes, the goal included.

In Appendix F, we explain further analysis as the trade-off between the amount of human querying and sample complexity of HuGE, as well as showing that HuGE is also robust to incomplete feedback and further details of its robustness to noisy underlying reward functions compared to PPO.

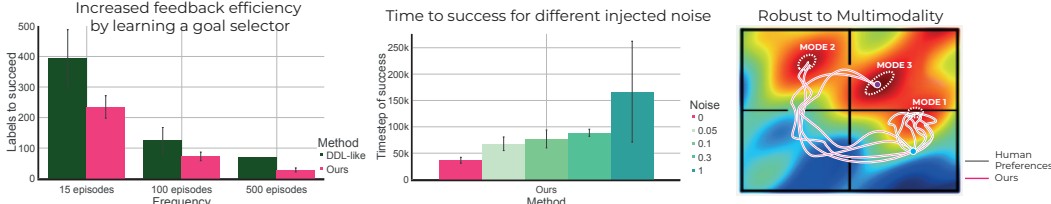

Figure 8: **(left)**: Learning a goal selector (*Ours*) needs on average 50% fewer labels than not (*DDL*) **(middle)**: As the noise in the feedback increases, so will the number of timesteps to succeed, however HuGE still finds a solution. **(right)**: HuGE is robust to noisy goal selectors since trajectories going to each mode will be sampled while if we ran RL the policy would become biased and fail. See Appendix F for more details.

## 6  Discussion

In this work, we build from the insight of decoupling human feedback from policy learning. We introduced HuGE, that guides exploration by leveraging small amounts of noisy and asynchronous human feedback, improving upon indiscriminate exploration or relying on hindsight generalization. Moreover, the policy training using self-supervised learning remains disentangled from the human feedback and makes it robust to asynchronous and noisy human feedback. We firmly believe this insight is key to learn policies from crowdsourced human feedback, where the noise in the labels due to non-expertise and variety of the annotators will be significant. As we show, when the annotations are noisy, optimizing over the learned noisy reward function using standard RL techniques will fail. However, once we disentangle the human feedback from the learned policy, and only use this to softly guide exploration, we can still learn optimal policies despite the noise. We rigorously demonstrate that HuGE is able to solve difficult exploration problems for various control tasks learning in both the real world and in simulation. Moreover, we run a crowdsourced data collection experiment with 109 human annotators living across three continents, with a wide variety of backgrounds, education level, ages, and we show HuGE succeeds in learning optimal policies. Finally, we provide ablations and analysis on the design decisions and better show the benefits of HuGE.

There are a number of directions for future work that are very exciting building on HuGE. On an immediate note, adapting HuGE to work in reset-free environments so that we can scale up the applications on learning on the real robot is the most promising. It is with excitement that we would like to see this system scaled to fine-tune foundation models in robotics. In the future, aiming to learn behaviors not just from bianry comparative feedback, but from richer forms of communication such as language or physical corrections or vector valued feedback would be intriguing as well.

# 7 Acknowledgements

We thank all of the participants to our human studies that gave us some of their time to provide labels. We thank the members of the Improbable AI Lab and the WEIRD Lab for their helpful feedback and insightful discussions. We also thank the reviewers for their insightful comments and helping us make the paper stronger.

The authors acknowledge the MIT SuperCloud and Lincoln Laboratory Supercomputing Center for providing HPC resources that have contributed to the research results reported within this paper. This research was supported by MIT-IBM Watson AI Lab.

# 8 Contributions

**Marcel Torne** and **Abhishek Gupta** jointly conceived the project. **Marcel** set up the simulation and training code, designed and conducted experiments in simulation, designed and implemented the interface for collecting human feedback, led the human experiments, conducted the ablations, and made the figures. **Marcel** led the manuscript writing together with **Abhishek**. **Max Balsells** designed and conducted the experiments in the real-world, integrated the vision models in the code, helped running some of the simulation experiments and ablations and helped writing the manuscript. **Zihan Wang** provided feedback on the manuscript. **Samedh Desai** helped **Max** setting up the real-world experiments. **Tao Chen** was involved in the initial research discussions and provided feedback on the paper. **Pulkit Agrawal** was involved in research discussions, contributed some of the main ideas behind the project and provided feedback on the writing and positioning of the work. **Abhishek Gupta** conceived the project jointly with **Marcel**, led the manuscript writing together with **Marcel**, and provided the main overall advising.

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

Next, we provide additional details of our work. More concretely:

- Appendix A **Real-Robot experiments**: provides more details on the real-robot experiments and on HuGE working from images.

- Appendix B **Real-Human experiments**: shows all the details of the real human experiments presenting results for both the crowdsourcing experiment as well as experiments on more benchmarks with fewer annotators.

- Appendix C **HuGE from demos**: we show HuGE is compatible with warm starting from demonstrations.

- Appendix D **Simulated Benchmarks**: we provide further details on the simulated benchmarks used in this work.

- Appendix E **Baselines**: we provide details about the baselines together with more detailed learning curves of the baselines on the simulated benchmarks.

- Appendix F **Further Analysis and Ablations**: we provide more insights on where the benefits of HuGE come from, as well as provide some ablations on the method.

- Appendix G **Implementation Details**: we provide further implementation details, hyperparameters and resources used.

The code is available at `github.com/Improbable-AI/human-guided-exploration`

## A    Real-Robot Experiments

HuGE's qualities make it suitable to learn policies directly in the real world. However, we adapted the method with respect to the simulated experiments in Fig 5. The main change consisted in changing the state space to image space instead of point space. Next, we show HuGE works from image space in two of the simulated environments, four rooms and block stacking.

### A.1    HuGE from images

Adapting HuGE to work from image space was a trivial process, the goal selector and policy networks were modified introducing a first encoding network consisting of 3 convolutional layers (with stride 2 and kernel size 5) to map the input image to a lower dimension space and then we passed it through an MLP to predict the score and action respectively.

**Learning stopping criteria**: When using the point state space, we could easily detect whether the policy stopped, indicating it reached the target goal, or that it got stuck, to then start random exploration from there. This could be done by computing the Euclidean distance and setting a small enough threshold. This is more difficult when working from image state space. What we did was train an image classifier $\phi(s_1, s_2)$ that predicts whether the two images correspond to states close in space (i.e. the state corresponding to image $s_2$ can be reached from the state corresponding to image $s_1$ within $t_{\text{close}}$ timesteps). We trained $\phi$ by using contrastive learning [15]. In particular, we sampled images from our replay buffer and assigned the corresponding label based on their distance in timesteps: $l(s_i, s_j) = 1$ if $|i - j| \leq t_{\text{close}}$ and 0 if $|i - j| \geq t_{\text{far}}$. Based on the premise that, in most cases, images obtained far away in time, will probably correspond to states that take longer than $t_{\text{close}}$ timesteps to reach, if we were to act optimally.

### A.2    Results in the real-world

For the real robot experiment, we used a LoCoBot with a WX-200 arm.

**Pick and place in the real world**: The state space consisted of RGB images of $64 \times 64$ pixels, and the action space was continuous with dimension 2, representing an absolute position in the space $(x, y)$ from which to predict a grasping point in even timesteps, or a dropping point in odd timesteps. For the experiment to be succeed in a reasonable amount of time, we pretrained the policy and the goal selector by using 5, sub-optimal demonstrations. The robot was trained for around 30h, during which, 130 labels were provided via the interface shown in B.4 by one annotator. Finally, we used a reset mechanism to pull the socks to the same corners, though, it had some stochasticity.

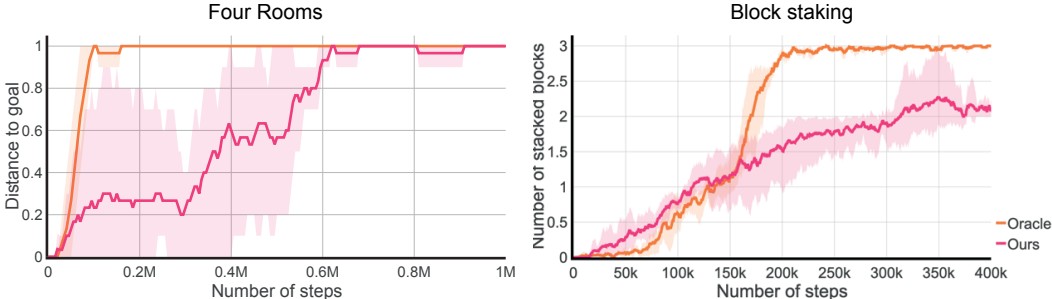

Figure A.1: Success rate for the four rooms (**left**) and block stacking (**right**) using images as input space for both the policy and goal selector.

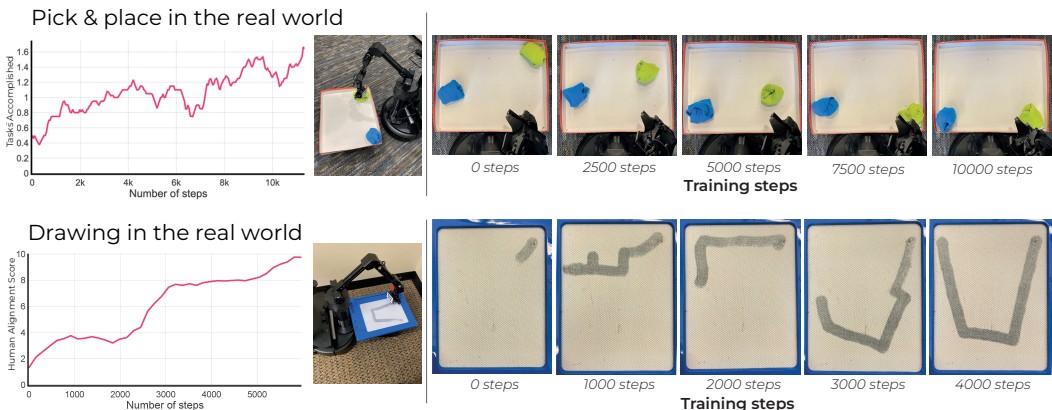

Figure A.2: Accomplished goals at the end of 5 different evaluation episodes along training on the real world to pick and place, and draw the letter U in the real world.

**Drawing in the real world**: The state space consisted of RGB images of $64 \times 64$ pixels, and the action space was discrete, encoding a total of 5 actions: no movement and moving across the two axis on the plane in polar coordinates (i.e. increasing $r$, decreasing $r$ and moving a fixed amount clockwise or counterclockwise), to move the end effector with the brush. An episode consisted of 12 timesteps. For the experiment to be ran in a reasonable amount of time, we pretrained the policy and the goal selector by using 5, sub-optimal demonstrations. The robot was trained for around 6h, during which, 150 labels were provided via the interface shown in B.4 by one annotator. Finally, the reset was done by using the erase mechanism in this drawing boards and moving it with the arm by using a script. As a final note, in this environment we had to perform few exploration steps and slowly increase the frontier. This is because in this environment there is only one optimal solution (the actions taken must be exactly the optimal ones, due to the fact that all past actions within the episode will affect the current state of the board), in particular, any non-optimal action will leave a trace, making that trajectory not that useful for the policy to learn from it.

**Human Alignment evaluation for drawing in the real world**: Designing a reward function for drawing is a hard and tedious labor. HuGE does not need a reward function and we can fully leverage human feedback to learn this behavior as shown in A.2. Without a reward function evaluation cannot be performed either. For this reason, we defined this "Human Alignment Score" that basically consists in querying humans and asking them for a score between 0 and 10 of how well the robot draw the target picture. In the case of the drawing experiments, we asked 2 annotators to label the performance of the robot drawing the letter U with a score from 0 to 10. This score was only used for evaluation and is the metric used to plot the drawing plot in A.2.

# B Real-Human experiments

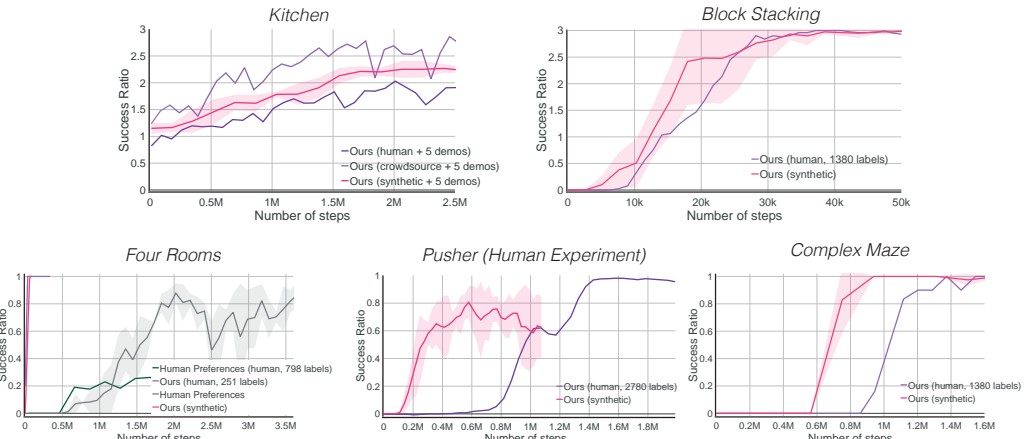

Figure B.3: Learning Progress with Human-in-the-Loop Feedback for the kitchen, block stacking, four rooms navigation, pusher walls and complex maze.

In this section, we give more details on how we ran the human experiment. We designed a simple interface shown in Figure B.4. We can see the two states to be compared in blue and red and the goal we aim to achieve in green. Then the annotator has to decide which one of the two states is closer to the given goal and provide feedback by clicking either on the blue or red button. In the case in which the annotator is undecided, they can click on the gray button that simply skips the current case. Finally, if the annotator does not provide any feedback after 30 seconds of being presented with the scenario, we skip the current batch of labeling and continue with training the policy. With this, we can take advantage of the properties of our method and continue training the policy even when no labels are given.

In Figure B.3 we share again the results obtained with the human experiment on a larger scale. We ran both experiments using the same frequency of labeling and number of labels per batch. In particular, We labeled every 50 rollout trajectories and queried the annotators for 20 labels. These parameters were identified through empirical experiments.

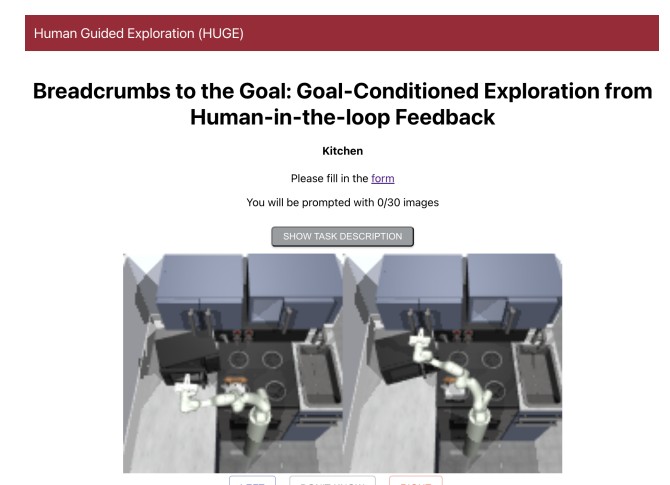

Figure B.4: Screenshot of the interface from our proposed crowdsourcing platform. It shows a comparison of two image states of the kitchen environment, and the user needs to click one of the three buttons below depending on their answer of which one is best: Left/I don't know/right

### B.1   Details about the crowdsourcing experiment

**Subject** 109 subjects participated in this pilot crowdsourcing study. Subjects were recruited from the acquaintances of the collaborators. The average time to complete the study was about 2 minutes. The subjects participated voluntarily without financial remuneration. The participants age ranged from 18 to 65+ years old. Gender Male=58.7%, Female=39.4%, Non-binary=1.8%. The participants were from 21 nationalities and participated from 13 distinct countries. More detailed information is presented in tables B.1 and B.2. There is no reason to believe that subjects experienced any physical or mental risks in the course of these studies.

**Procedure** This study was approved by the Institutional Review Board of the Massachusetts Institute of Technology protocol E-4967.

We provide the participants with the following detailed instructions:

> Thank you very much for participating in this study. It should not take you more than a couple of minutes to complete. First of all click on the link we sent you to get directed to the main page (B.4), you can either use your phone or your computer. Please, start by filling out the form for us to get an overview of the participants' demographic. The task consists of controlling a robot to do different things in the kitchen: 1) open the slider on the right, 2) open the microwave on the right 3) open the hinge cabinet on the left. [We show a video of a successful trajectory]. To help us, we will present you two images and you need to tell us which one of the two images is closer to achieving the task. Click on the left/right button depending on whether the left/right image is better. If at some point you don't know which one is best please click the "I don't know" button. [We present a couple of examples demonstrating this]. We will show 30 pairs of images and after that, you will receive a message saying you completed the task. You can stop at any moment before that if you want.

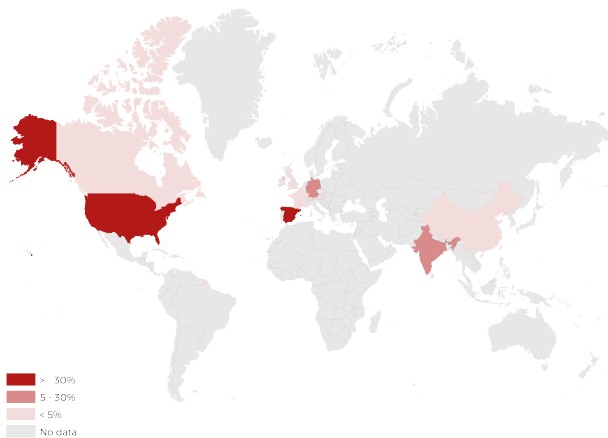

Figure B.5: Heatmap on the country representation during our crowdsourcing experiment.

| Metric | Percentage |
|---|---|
| **Current country of Residence** | |
| USA | 41.3% (45) |
| Spain | 30.3% (33) |
| India | 8.3% (9) |
| Germany | 6.4% (7) |
| Canada | 2.8% (3) |
| France | 2.8% (3) |
| Singapore | 1.8% (2) |
| China | 1.8% (2) |
| Andorra | 0.9% (1) |
| Austria | 0.9% (1) |
| Ireland | 0.9% (1) |
| Switzerland | 0.9% (1) |
| United Kingdom | 0.9% (1) |
| Prefer not to say | 0% (0) |
| **Gender** | |
| Male | 58.7% (64) |
| Female | 39.4% (43) |
| Non-binary | 1.8% (2) |
| Prefer not to answer | 0% (0) |
| **Age group** | |
| 18-24 | 48.6% (53) |
| 25-34 | 24.8% (27) |
| 35-44 | 7.3% (8) |
| 45-54 | 11.0% (12) |
| 55-64 | 7.3% (8) |
| 65+ | 0.9% (1) |
| Prefer not to answer | 0% (0) |
| **Education** | |
| Graduate or professional degree | 39.4% (43) |
| College degree | 33.9% (37) |
| High school or some college | 20.2% (22) |
| Other | 12.8% (14)) |
| Prefer not to say | 2.8% (3) |

Table B.1: Demographics on the participants of the crowdsourced data collection experiment

| Metric | Percentage |
|---|---|
| **Nationality** | |
| Spain | 26.6% (29) |
| USA | 20.2% (22) |
| India | 9.2% (10) |
| Germany | 8.3% (9) |
| China | 7.3% (8) |
| France | 4.6% (5) |
| Mexico | 3.7% (4) |
| Colombia | 2.8% (3) |
| Switzerland | 1.8% (2) |
| Hong Kong | 1.8% (2) |
| Canada | 1.8% (2) |
| Uruguay | 0.9% (1) |
| Singapore | 0.9% (1) |
| Russia | 0.9% (1) |
| Ireland | 0.9% (1) |
| Lebanon | 0.9% (1) |
| South Korea | 0.9% (1) |
| Sweden | 0.9% (1) |
| Andorra | 0.9% (1) |
| Puerto rico | 0.9% (1) |
| Israel | 0.9% (1) |
| Prefer not to say | 0.9% (1) |
| **Ethnicity** | |
| Hispanic, Latino or Spanish | 38.5% (42) |
| Asian | 28.4% (31) |
| White or Caucasian | 24.5% (27) |
| Middle Eastern or North African | 3.7% (4) |
| South-east Asian | 2.8% (3) |
| Black or African American | 0.9% (1) |
| Perfer not to say | 0.9% (1) |

Table B.2: Demographics on the participants of the crowdsourced data collection experiment

# C  HuGE from demonstrations

As explained in Section 4.3, HuGE is suited to easily incorporate demonstrations and therefore use these to warm start the method providing a significant reduction in sample complexity over HuGE without demonstrations. Below we show a comparison of HuGE against other baselines that use demonstrations.

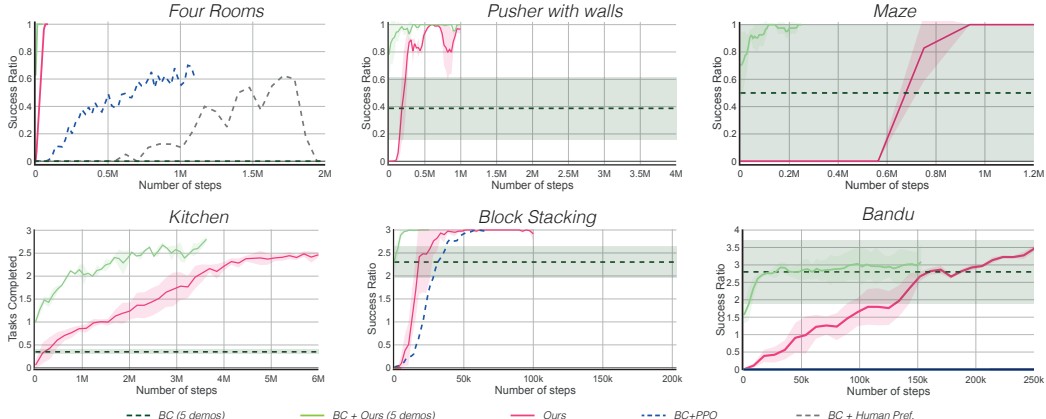

Figure C.6: Comparison of HuGE warmstarted with demonstrations. We observe that a significant speed up is provided compared to running HuGE without demonstrations. Furthermore, we observe an improvement over simply learning from demonstrations (Behavior Cloning, BC) baselines. Similarly, HuGE with demonstrations continues to outperform the PPO and Human Preferences with demonstrations baselines.

# D    Simulated Benchmarks

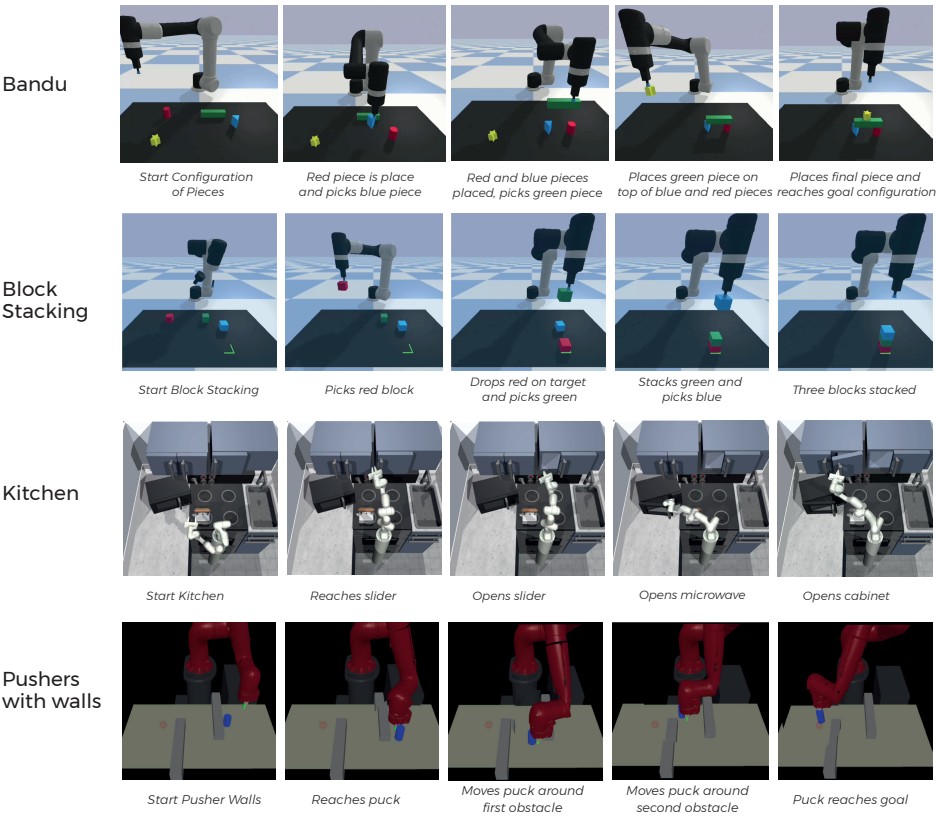

Figure D.7: Results of our method on four of the hardest benchmarks. From left to right, the timestep in the trajectory increases.

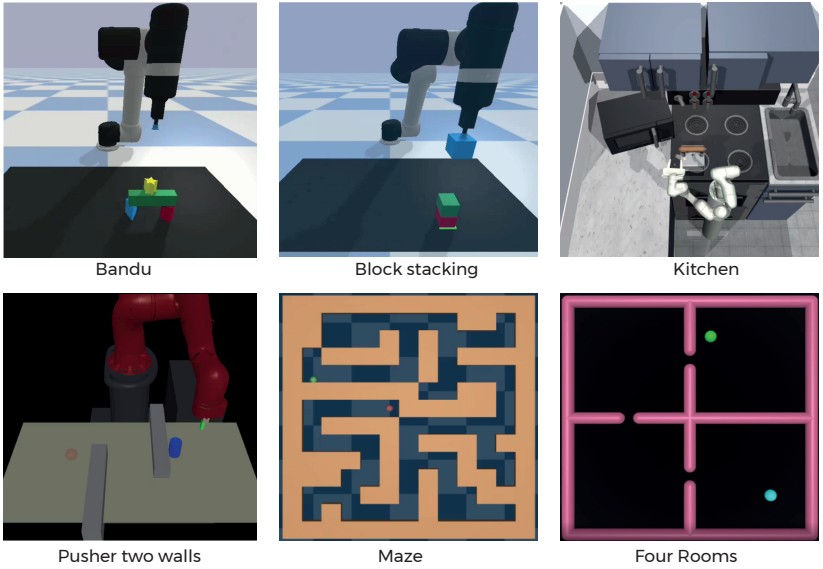

Figure D.8: Results of our method on four of the hardest benchmarks. From left to right, the timestep in the trajectory increases.

In this section, we give more details on the benchmarks used to compare our method with the baselines. All of these benchmarks are variations of benchmarks presented in previous work. In general, we have made them harder to showcase the benefits of our method. More concretely, for each method, we will give an overview of the difficulties it has and we will present the reward function we designed to provide synthetic labels in our experiments.

1. **Four rooms (small 2D Navigation)**: We consider goal-reaching problems in a 2-D navigation problem in a four rooms environment, shown in Fig D.8. The challenge in this environment is navigation through obstacles, which are unknown without exploration. The agent is initialized in the bottom right room and the goal is sampled from the top right room. The state observation of this environment is the absolute position of the agent in the world, i.e. a vector $(x, y)$, and the action space is discrete with 9 possible actions, encoding 8 directions of movement (parallel to the axis and diagonally), plus a non-move action. To solve this benchmark the agent needs to traverse the two intermediate rooms to get to the target room, traversing a total of four rooms.

   The reward function in this case is the shaped distance between the state and the goal. This benchmark is a modification of the benchmarks proposed by [24].

2. **Maze (large 2D Maze Navigation)**: We consider a second 2-D navigation problem in a maze environment. The additional challenge in this environment compared to the previous one relies upon having a longer horizon (see Figure G.4). The agent is initialized in the green dot and has to reach the red dot. The state space is the absolute position of the agent in the maze, i.e. a vector $(x, y)$, and the action space is the same as in the Four rooms one, i.e. discrete with dimension 9.

   The reward function in this case is the shaped distance between the state and the goal.

3. **Pusher two walls**: This is a robotic manipulation problem, where a Sawyer robotic arm pushes an obstacle in an environment with multiple obstacles. The puck and arm start in the configuration seen in Fig D.8. The task is considered successful if the robotic manipulator brings the puck to the goal area, marked with a red dot.

   The state space of this environment consists of the position of the puck and the position of the arm. The action space is the control of the position of the robotic arm. It is also a 9-dimensional discrete action space where each one corresponds to a delta change in the position in 2D. This benchmark is a modification of the benchmarks proposed by [24]. The reward function designed for this environment is the following:

   $$r = max(distance\_puck\_finger, 0.05) + distance\_puck\_goal$$

4. **Sequential Kitchen Manipulation**: This benchmark is a harder robotic manipulation task where apart from being long horizon the agent needs to show three different skills to solve the task. We operate a 7 DoF Franka robot arm in a simulated kitchen, manipulating different cabinets, sliding doors, and other elements of the scene. The observation space consists of the position of the end effector and its rotation together with the joint states of the target objects. The action space consists in controlling the end effector position in 3D, we discretize it so the dimension is 27, and the control of the gripper and rotation of the arm. In our evaluation, we consider tasks where the goal is to sequentially manipulate three elements in the kitchen environment - the sliding cabinet, the microwave and the hinge cabinet to target configurations. The reward function we use is the following:

$$r = \begin{cases} -\text{distance(arm, hinge cabinet)} - |\text{hinge cabinet target joint - hinge cabinet current joint}| \text{ , if slide cabinet and microwave opened} \\ -\text{distance(arm, microwave hinge)} - |\text{microwave target joint - microwave current joint}| - \text{bonus , if slide cabinet opened} \\ -\text{distance(arm, slide cabinet hinge)} - |\text{slide cabinet target joint - slide cabinet current joint}| - 2\text{bonus , otherwise} \end{cases}$$

$$(5)$$

5. **Block Stacking**: This domain is another long horizon robotic manipulation task, we operate a 6 DoF UR5 robot arm with a suction gripper as an end effector in a simulated tabletop configuration, stacking blocks. The observation space consists of the position of the end effector and the position of each block in 2D, and a bit indicating whether the hand is holding a block. This is a continuous action space domain with dimension 2, where the agent will predict a grasp position if it does not hold an object and a drop position if it is holding an object. We consider the goal to be accomplished if the three blocs are stacked in the correct

order (red, green, blue) on the correct fixed place on the table. Our code was inspired from the Ravens benchmark [57]. The reward function is the following:

$$r = - \begin{cases} \text{distance(arm, blue block) + distance(blue block, target goal) , if red and green block at position} \\ \text{distance(arm, green block) + distance(green block, target goal) + bonus , if red block at position} \\ \text{distance(arm, red block) + distance(red block, target goal) + 2bonus , otherwise} \end{cases}$$

(6)

$$r = - \begin{cases} \text{d(arm, blue) + d(blue, goal) , if red and green at goal} \\ \text{d(arm, green) + d(green, goal) + bonus , if red at goal} \\ \text{d(arm, red) + d(red, goal) + 2bonus , otherwise} \end{cases}$$

(7)

6. **Bandu**: This domain is very similar to the block stacking. We operate a 6 DoF UR5 robot arm with a suction gripper as an end effector in a simulated tabletop configuration. The observation space consists of the position of the end effector and the position of each block in 2D, and a bit indicating whether the hand is holding a block. This is a continuous action space domain with dimension 2, where the agent will predict a grasp position if it does not hold an object and a drop position if it is holding an object. We consider the goal to be accomplished if the four blocs are stacked in the target configuration building the castle like structure seen in Figure D.8. Our code was inspired from the Ravens benchmark [57]. The reward function is the following:

$$r = \begin{cases} -\text{distance(arm, yellow star) - distance(yellow star, target yellow star) , if all except star at position} \\ -\text{distance(arm, green block) - distance(blue green block, target green block)} - \text{bonus , if red and blue blocks at position} \\ -\text{distance(arm, blue triangle) - distance(blue triangle, target blue triangle)} - 2\text{bonus , if red cylinder at position} \\ -\text{distance(arm, red cylinder) - distance(red cylinder, target red cylinder)} - 3\text{bonus , otherwise} \end{cases}$$

(8)

More details about how these benchmarks were run, such as the number of episodes we ran the benchmarks for, are presented in Section G

# E Baselines

We compare HuGE to relevant baselines from prior work.

1. **Inverse Models:** We compare with the iterative supervised learning algorithm for goal-reaching introduced in [24], consisting of hindsight relabeling without additional exploration (GCSL).

2. **Learning from Human Preferences:** We consider the technique introduced in [16], which learns a goal-agnostic reward model using binary cross-entropy. This learned reward is then combined with an on-policy RL algorithm [52] to learn the policy.

3. **DDL:** Dynamical Distance Learning [27] proposes a method to learn a goal-conditioned reward function by regressing on the time distance between states achieved in the same trajectory. A human synchronously provides preferences on which state brings the agent closest to the goal, note that no goal selector is being learned. The policy is then trained to maximize the learned reward to get to this selected state.

4. **Go-Explore/LEXA:** We compared with a version of goal-reaching with indiscriminate exploration. In particular, we perform frontier goal selection by identifying goals with the lowest densities. The policy returns to these states and perform random exploration from there. This is equivalent to performing indiscriminate exploration.

5. **Proximal Policy Optimization:** We compare with an on-policy algorithm [52] with both a standard sparse and dense reward to directly optimize the goal-reaching objective.

6. **Behavior Cloning**: Supervised learning on a batch of expert trajectories. In our experiments we use 5 expert trajectories.

7. **Behavior Cloning + Ours**: We pretrain the policy using imitation learning and we warm start our goal selector by training it from the expert trajectories. Given two random states in the same expert trajectory we add them into the training data for the goal selector, setting the state further in time as closest to the goal.

These baselines are chosen to compare HuGE with methods that perform pure exploration, hindsight relabeling, and human preferences without being goal conditioned to highlight the benefits of combining goal-driven self-supervision with human-in-the-loop exploration guidance.

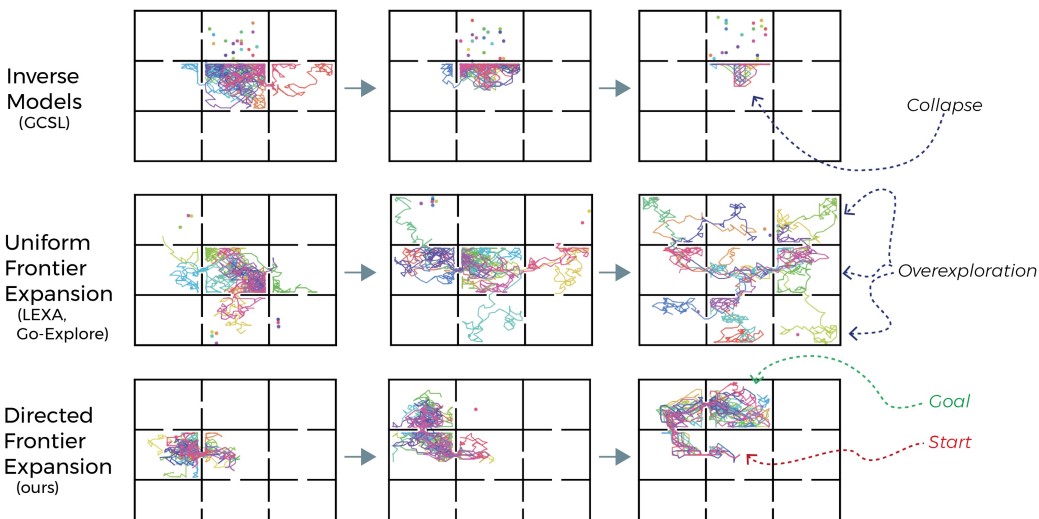

Figure E.9: Failure modes of exploration algorithms for goal-reaching. Inverse models (top) collapses and does not discover the target room (second room at the top). Uniform frontier expansion (middle) does reach the target room, but to get there it visits all possible rooms, since exploration is indiscriminate. Directed frontier expansion (bottom, ours) reaches the target room much faster by leveraging human signal on direction. Training epochs increase from left to right. Each subfigure is an aerial view of a floor with 9 rooms, with multiple trajectories, each one in a different color.

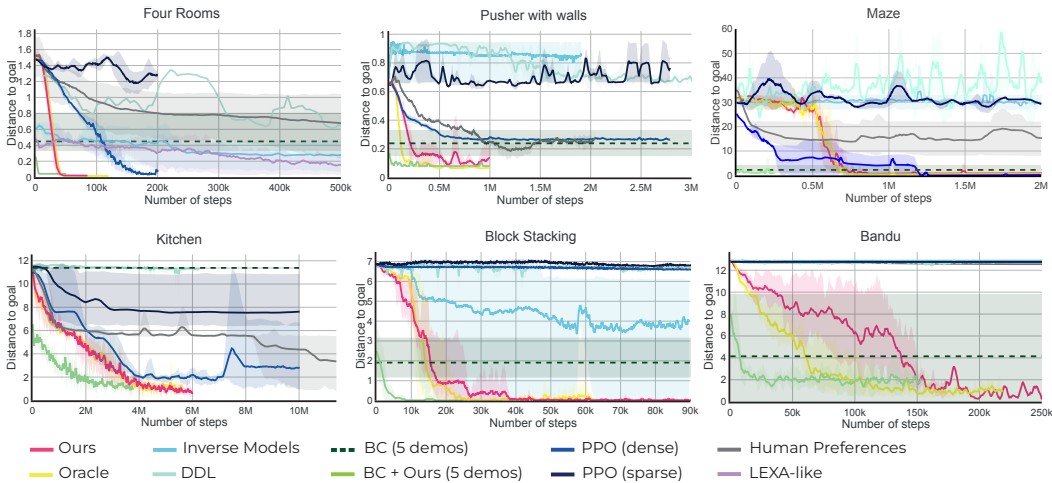

Figure E.10: Distance to the goal for each method on different benchmarks. We note that the LEXA-like exploration strategy was only implemented on the four rooms benchmark.

For the sake of concreteness, we will study two simple schemes from prior work on solving goal-reaching problems —self-supervision via goal conditioned supervised learning [24] (as described in Section 3) and reinforcement learning with density based exploration [37]. Exploration in GCSL relies on generalization of the policy across goals, while density based exploration rewards exploring the most novel states. We show these algorithms can fail in different ways for a simple maze environment shown in Fig E.9, where the agent starts in the middle room and must reach goals commanded in the top middle room.

As shown in Fig E.9, GCSL exploration quickly collapses in the maze environment. This can be understood by noticing that self-supervised training on goals in the bottom right corner room or even the bottom left corner room does not extrapolate to the top right corner, where the commanded goals are. Instead of navigating the agent around the walls, the policy generalization suggests that the agent simply go into the wall as shown in Fig E.9.

Exploration methods are meant to tackle this kind of degenerate exploration, by encouraging visitation of less frequently visited goals at the "frontier" of visited states. When applied to the goal-reaching problem, in Fig E.9, we see that while the exploration is not degenerate, exploration is indiscriminate in that it explores both sides of the maze even though commanded goals are only down one particular path. While this will eventually succeed, it incurs a significant cost of redundant exploration by going down *redundant* paths.

This suggests that frontier expansion is needed like exploration methods, but should ideally be done in a directed way towards goals of interest. In Figure E.9 we see how this directed exploration could be useful and reduce sample complexity, by removing the need for indiscriminate frontier expansion. We show how a small amount of relatively cheap human feedback can be leveraged to guide this exploration.

## E.1 Detailed training curves

For some of the runs the plot of the success could be misleading, in the sense that, despite not achieving the goal, the algorithms may still learn how to almost solve the task, or at least gained some knowledge about how to approach it. Figure E.10 shows for each of the runs, the distance to the goal, which corresponds to $-r$ where $r$ is the reward of the corresponding benchmark, as described in Section D.

For example, by looking at Figure E.10, we can see that despite the fact that the Human Preferences wasn't able to complete some of the tasks, such as Four Rooms, Pusher with walls or Maze, it still got some insight on how to approach it, getting much closer to the goal than the other methods that failed.

| Benchmark | Oracle | Ours | GCSL | Human Preferences | DDL | PPO (sparse) | PPO (dense) | GCSL + Go-Explore |
|---|---|---|---|---|---|---|---|---|
| 4 rooms | **0.02 ± 0.01** | **0.02 ± 0.00** | 1.15 ± 0.67 | 0.48 ± 0.39 | 0.45 ± 0.28 | 1.45 ± 0.13 | **0.05 ± 0.02** | 0.13 ± 0.11 |
| Maze | **0.4 ± 0.3** | **0.8 ± 0.3** | 29.6 ± 2.2 | 18.5 ± 5.6 | 8.54 ± 10.6 | 30.4 ± 0.7 | **0.0 ± 0.2** | 1.1 ± 0.14 |
| Pusher | **0.06 ± 0.00** | **0.11 ± 0.04** | 0.85 ± 0.11 | 0.26 ± 0.03 | 0.69 ± 0.06 | 0.72 ± 0.06 | 0.27 ± 0.00 | 0.50 ± 0.29 |
| Kitchen | **1.06 ± 0.32** | **0.67 ± 0.21** | 11.72±0.11 | 3.43 ± 4.37 | 11.28±0.02 | 7.63 ± 4.96 | 2.84 ± 2.72 | 10.97 ± 0.13 |
| Stacking | **0.1 ± 0.2** | **0.0 ± 0.0** | 4.1 ± 2.3 | 6.5 ± 0.1 | 6.6 ± 0.1 | 6.7 ± 0.0 | 6.6 ± 0.0 | 3.76 ± 2.66 |
| Bandu | 1.00 ± 0.53 | **0.36 ± 0.73** | 12.87±0.01 | 12.54±0.01 | 12.63±0.21 | 12.75±0.01 | 12.75±0.01 | 10.7 ± 1.51 |

| Benchmark | PEBBLE | BC (5 demos) | BC + Ours (5 demos) |
|---|---|---|---|
| 4 rooms | 0.59 ± 0.1 | 0.45 ± 0.46 | 0.04 ± 0.00 |
| Maze | 11.95 ± 2.56 | 2.25 ± 1.51 | 0.87 ± 1.02 |
| Pusher | 0.185 ± 0.04 | 0.25 ± 0.09 | 0.08 ± 0.01 |
| Kitchen | 10.48 ± 0.01 | 11.38 ± 0.00 | 0.87 ± 1.02 |
| Stacking | 4.24 ± 1.51 | 1.91 ± 1.02 | 0.01 ± 0.00 |
| Bandu | 13.5 ± 3.25 | 4.21 ± 5.47 | 1.87 ± 0.4 |

Figure E.11: Average distance and standard deviation across 4 seeds for the different baselines we implemented to compare against HugRL. We see that HugRL consistently succeeds (in bold) to solve all benchmarks when most other baselines do not. The oracle would be the upper bound that we could hope to achieve, since in this case labels are provided all the time, and the goal selector is substituted by a precise distance function.

## E.2 Analysis of PPO vs HuGE

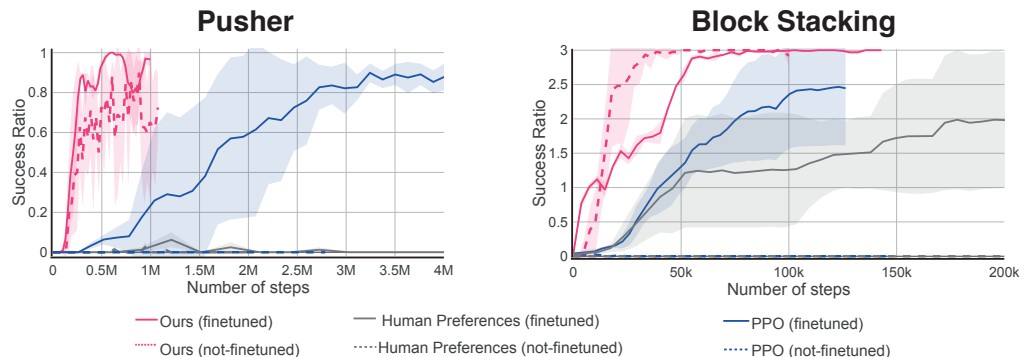

Figure E.12: Comparison of the PPO and Human Preferences baselines against HuGE when using a carefully tuned reward function and not on the pusher and block stacking environments.

As we have shown throughout this manuscript, HuGE is more robust to underlying noisy human preferences, which means it's robust to noisy underlying reward functions. In Figure E.12 we provide a comparison between the performances of PPO and HuGE when the reward is finetuned against when it's not. We observe that while HuGE succeeds in both cases, PPO only succeeds when the reward function is carefully tuned.

In the pusher experiment, the reward finetuning consisted in doing some hyperparameter tuning, basically, modifying the position of the keypoints and adding a bonus for solving each keypoint. In the block stacking, it consisted in adding an extra reward term seen in Figure E.13.

Finally, we wanted to understand why even when the reward function is finetuned and both HuGE and PPO obtain optimal policies, why is HuGE more sample efficient than PPO. To analyze this we ran a comparison on gradient variance using HUGE as compared to gradient variance with PPO, comparing

```python
def standard_distance(obj_pos, goal_pos):
    for i in range(num_blocks):
        if norm(obj_pos[i] - goal_pos) < 0.1:
            continue

        return (
            norm(arm_pos - obj_pos[i])
            + norm(obj_pos[i] - goal_pos)
            + bonus * (self.num_blocks - i - 1)
        )
    return 0
```

```python
def finetuned_distance(obj_pos, goal_pos):
    for i in range(num_blocks):
        if norm(obj_pos[i] - goal_pos) <= 0.1:
            continue

        if norm(obj_pos3 - ee_pos) > 0.05:
            return (
                norm(obj_pos[i] - ee_pos)
                + 2 * bonus * (num_blocks - i)
            )

        return (
            norm(arm_pos - obj_pos[i])
            + norm(obj_pos[i] - goal_pos)
            + bonus * (2 * (num_blocks - i) - 1)
        )
    return 0
```

Figure E.13: Comparison of the original non-finetuned reward function (left) against the finetuned one (right) for the block stacking experiment.

the average pairwise cosine similarity across mini-batches to measure gradient variance (as suggested in In Figure E.14). We find that training the gradient similarity is significantly larger in HuGE than in PPO, potentially explaining improved sample complexity. Besides this, other components in HuGE may also lead to faster learning as generalization across states that come from self-supervised hindsight relabeling and hindsight relabeled policy learning is off-policy while PPO is on-policy. However, we donot want to claim that the synthetic variant of HuGE is fundamentally a better optimization algorithm than PPO. Given a carefully shaped reward function, PPO typically eventually works on most decision-making problems, however, HuGE is a method for goal-reaching problems, which is a subset of all possible decision-making problems. In these goal-reaching problems, we have shown that HuGE can typically outperform PPO even at convergence when the reward functions are noisy. However, PPO is a more general learning paradigm and more careful experimentation and ablations are needed before making broad-reaching claims.

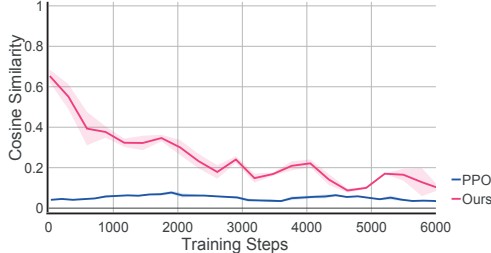

Figure E.14: PPO vs HuGE similarity of gradients inside a minibatch during training. We observer HuGE's similarity is much larger, which implies the gradient variance is lower. This could mean that the optimization shape is smoother for HuGE and could be one of the reasons why HuGE outperforms PPO.

### E.3   Analysis of HuGE vs Human Preferences when learning from noisy feedback

We ran an additional experiment shown in Figure E.15 to investigate further the benefits of HuGE when compared to baselines that run RL algorithms over the learned reward functions. In this case we analyze the performance of both HuGE and Human Preferences, when providing noisy synthetic feedback and we observe that while HuGE succeeds in learning an optimal policy, Human Preferences does not.

## HuGE vs HP when learning from noisy feedback

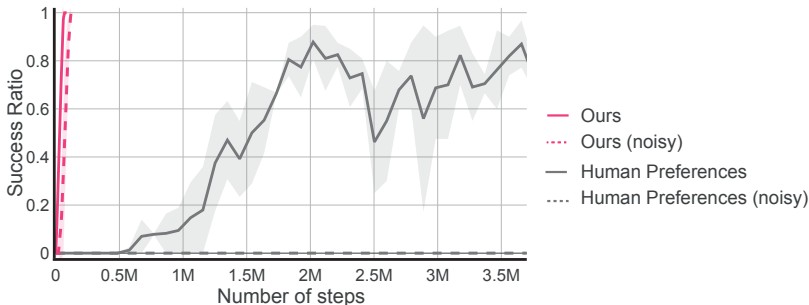

Figure E.15: We observe that when adding synthetic noisy feedback HuGE converges to optimality while Human Preference fails

# F Further Analysis and Ablations

## F.1 Analysis on learning from comparisons

**There is a tradeoff between the frequency of labelling and the speed for the policy to converge.**
In Figure F.16, (**left**) we observe that if we query more frequently, the policy needs more labels to succeed, however, we also observe (**right**) that when querying less frequently it takes more timesteps to succeed. Meaning that if we provide labels more frequently, the policy is going to converge faster to the optimal policy, but will come at the cost of needing more human annotations. On the other hand, if the human annotators provide labels less frequently, it will take longer for the policy to converge to the optimal policy. The query frequency will hence be an important parameter to look into depending on what we want to optimize for, number of human labels or timesteps to succeed. We believe that for simulation experiments, we might want to optimize for using less human labels since the policy rollouts can be done very fast. However, when working with learning on the real robot, we might prefer to have humans label more frequently and reduce the number of rollouts in the real world, which is usually the bottleneck.

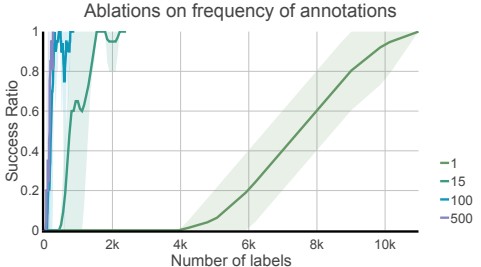
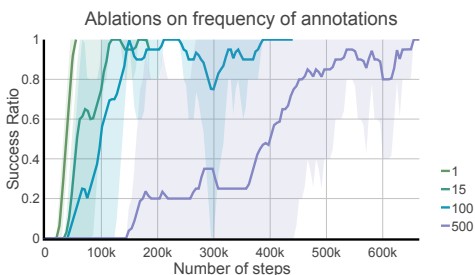

Figure F.16: On the left/right we show the number of labels/timesteps needed to succeed when varying the query frequency. 1, 15, 100, and 500 are the number of episodes between each period of querying the human for annotations. We observe a clear tradeoff between needing fewer labels to succeed against needing more timesteps. Meaning that if we query more frequently, we will need fewer timesteps to succeed and vice versa. These experiments are done in the Four Rooms benchmark.

**Querying a few samples per batch is enough.** In Figure F.17, (**left**) we observe that providing more labels every time we query the human leads to needing more labels to have successful policies, as expected. In **right**, however, we observe that the number of timesteps needed to have a successful policy is very similar when querying for 5,20 or 100 annotations, however, when only querying for 1 the performance drops significantly. This means that 5 labels are already enough to learn how to expand the frontier, and querying more than 5 labels brings useless information.

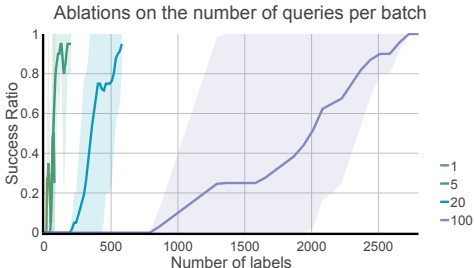
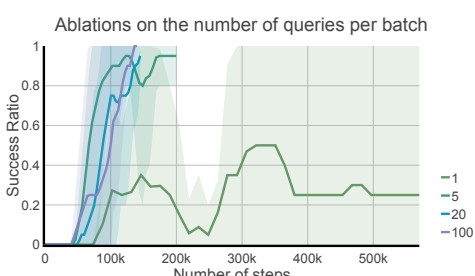

Figure F.17: On the right we show the number of steps needed to succeed in the four rooms benchmark depending on the number of comparisons queried per batch. On the left, we show the number of labels needed to succeed, again depending on the query batch size. We observe that we can go as low as 5 queries per batch, and the performance is similar to 20 and 100. Showing that too many queries bring duplicated information to the goal selector training. Also, we see that providing 1 label is not enough, degrading the performance significantly. These experiments are done in the Four Rooms benchmark.

**HuGE is robust to noisy labels.** Increasing the noise in human labels leads to an increase in the number of timesteps needed to for the policy to learn to achieve the goal, as seen in Figure F.19. However, this does not decrease the accuracy of the resulting policy. Increased noise in the labels makes exploration become less directed and closer to the uniform frontier expansion methods.

Having a closer look in Figure F.18 at the shape of the reward function when large noise is added to the feedback. We observe that the goal selector becomes less accurate compared to the one with perfect feedback in Fig F.23. However, HuGE still successfully reaches the goal. As we can see, there are 3 modes in the final step (4th subfigure in F.18). This means, the goals will be sampled most frequently from these 3 modes, which will result in a less efficient frontier expansion, since only one of the three modes is the target goal. However, since we are learning a goal-conditioned policy through self-supervised learning this remains unaffected by this noise and will learn to go to the three modes, one of which is our target location. This would not be the case for methods that use this goal selector as a reward function to run model-free RL, due to its convergence to local maxima without reaching the target goal.

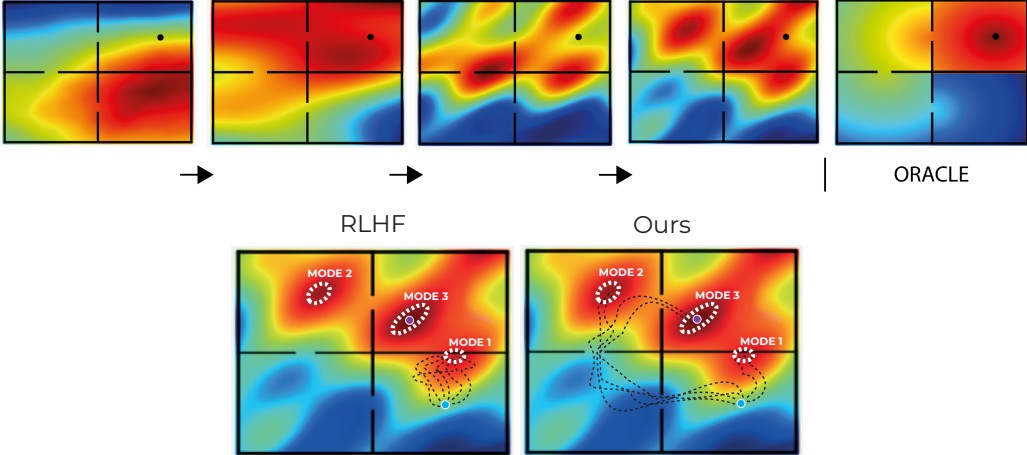

Figure F.18: Evolution of the learned goal selector when the distance for the synthetic human has a noise of 1. We observe that the goal selector is not accurate, however, our method still successfully reaches the goal, hence, it is robust to inaccurate goal selectors. This would not be the case for methods that use this goal selector as a reward function to run model-free RL, due to the noise on it and multiple local minimas and maximas.

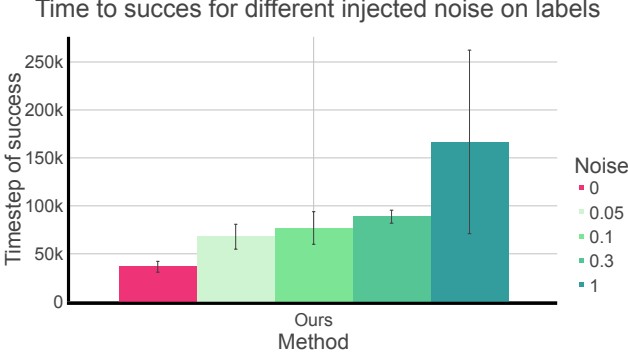

Figure F.19: Show the effect of adding Gaussian noise in the labels provided by the human on the Four Rooms benchmark. We observe that our method is robust to different amounts of added noise, however, as noise increases, so will the timesteps needed to succeed. Noise is injected into the distance function used by the synthetic human to provide labels, which means that with higher noise the probability of the comparison being wrong will increase. For context, the distance between the initial state and the goal is around $1.6$.

**HuGE is robust to underlying simple reward functions.** In Figure F.20 we show the performance of our method in the Four Rooms environment when dealing with a simplified version of feedback. In particular, we only return feedback if the given queried states have a distance difference of at least d with respect to the goal. For context, in this environment $0.5$ is approximately the distance between the center of two consecutive rooms, so using $d \geq 0.5$ is roughly similar to using the room number as a reward function. Therefore, in this experiment, we can see that, even with very simple reward functions, we can still get some insight on how to solve the task, though at the expense of clearly slower convergence. In particular, we can see how coarser reward functions lead to worse performances. This also helps us understand what happens in scenarios in which it is hard for humans to compare states that are similarly good for the purpose of achieving the required goal.

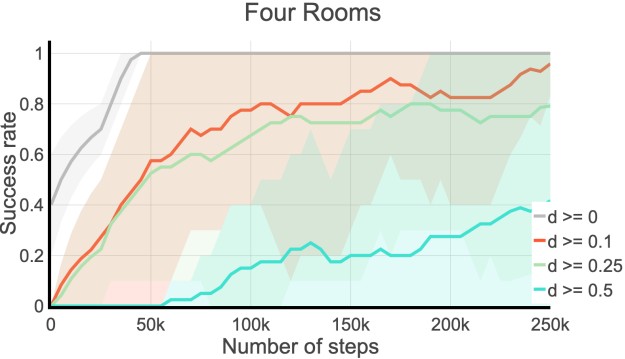

Figure F.20: Comparison on the effect of simplified reward functions providing the synthetic human annotations.

**HuGE can learn when no labels are provided.** This property of HuGE is because of the self-supervised learning used to train the policy but also a result of using a parametric goal selector as compared to directly selecting goals of interest as done in [27], which will not have this advantage. From Figure F.23 we observe that a parametric goal selector has the capacity to generalize while, by definition a non-parametric goal selection [27] will not. Thereafter, using a parametric reward model that has non-degenerate extrapolation can lead to significantly more frontier expansion. In Figure F.21 we show how our method succeeds in reaching the final goal room even if the goal selector has stopped training when the agent enters any of the previous rooms. However, this comes at a cost in much slower convergence.

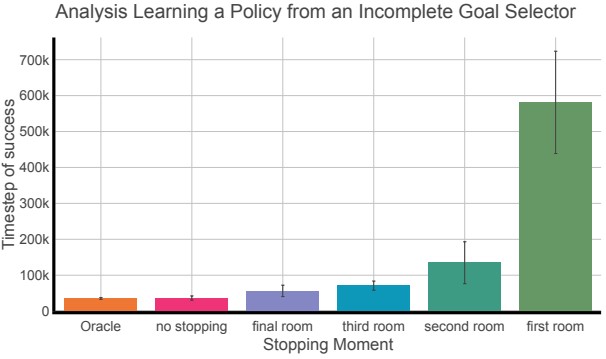

Figure F.21: Effect of freezing the goal selector at different points in the learning of the policy on how long it takes to learn a successful policy on the Four Rooms benchmark. We see that an earlier stop in the training leads to an increase in the timesteps needed to succeed. However, even if we stop in the second room, our method is still very good at quickly finding a successful policy, which shows how robust it is against incomplete goal selectors. This would not be the case for methods that run RL on the learned reward functions (as DDL, and RL from Human Preferences). The policy still succeeds thanks to the added random exploration, the self-supervised nature of GCSL, and a small probability of sampling the final goal.

## F.2 Goal selector Analysis

**Learning a goal selector is more feedback efficient than directly using the human feedback.**
In figure F.22 we show a comparison of the number of labels needed to succeed when using a parametric goal selector (Ours) against directly using the goal selected by the human (DDL). We show the comparison between different frequencies of human querying. 15, 100, 500 episodes are the number of episodes we wait before querying the human annotator for more labels. We observe that when learning a goal selector, we obtain a reduction in the number of labels needed of 40% when querying every 15 or 100 episodes and a reduction of 59% when querying every 500 episodes. Furthermore, if we don't learn this parametric model, with low frequencies we might not learn a successful policy, as happens for the non-parametric version at 100, 500 episodes of frequency. When using the non-parametric goal selector (DDL) not all trials succeed, for querying every 100 episodes, 2 seeds out of 4 fail and for 500 episodes between querying 3 out of the 4 fail, which is another reason why parametric goal selectors are better.

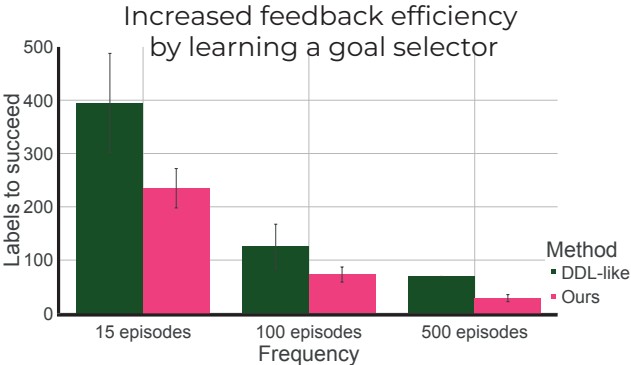

Figure F.22: Comparison of the number of labels needed to succeed when using a parametric goal selector (Ours) against directly using the goal selected by the human (DDL).

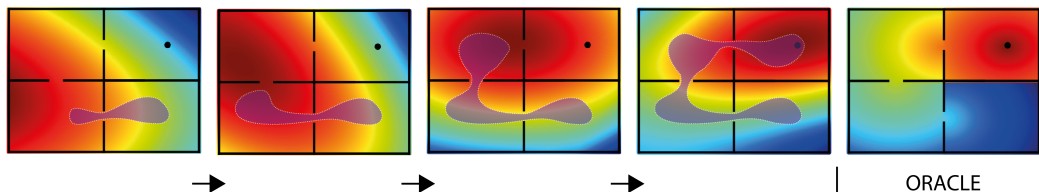

Figure F.23: Progress of goal selector learning in the four rooms environment as learning progresses it gets closer to the target (oracle on the right). The purple area represents the visited states by the agent at that point. We observe that the goal selector provides extrapolation which will help the training with fewer annotations.

In figure F.23, we show the goal selector will have non-trivial generalization, allowing us to continue expanding the frontier even when no human is present.

Furthermore, in F.24 we explore how accurate the goal selector is, depending on the number of queries it has been trained with. In particular, we tested it in the Four Rooms environment by training the goal selector using pairs of states sampled uniformly. During evaluation, given two states which are less than $d$ units apart, we compute the accuracy for which the model is able to pick the closest state to the goal. This allows us to see that the model is able to, given two states, determine which one is the closest to the goal, even when the given states are very close together and even when trained with just a handful of queries. For context, bear in mind that the distance from the initial state to the goal is 1.6 units.

**Qualitative analysis of the generalization of the goal selector.** In this qualitative analysis, we show visualizations of the learned goal selector as different rooms are discovered during the learning

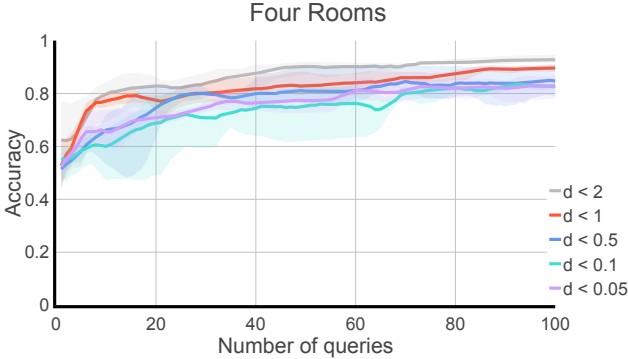

Figure F.24: Accuracy of the goal selector depending on the number of queries and dependent on the distance $d$ between the states compared in the labels.

process in the four-rooms domain Fig F.25. The goal selector model shows nontrivial extrapolation and can potentially provide guidance even beyond the set of states it is trained on.

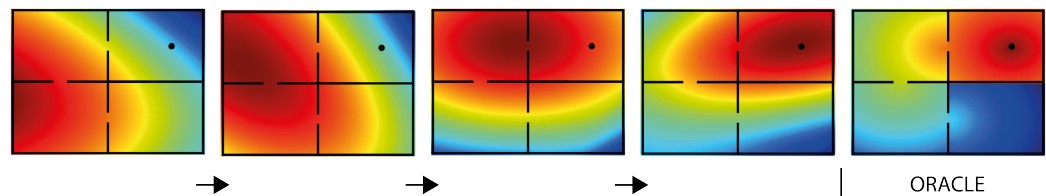

Figure F.25: The goal selector learns and converges to a result close to the oracle (rightmost) as epochs increase (left to right). We observe how this goal selector gets updated iteratively as the frontier expands. Colder colors mean a lower reward for that state, whereas warmer colors mean a higher reward for that state, in this case, this is equivalent to the distance to the goal.

## F.3  Compatibility of HuGE with Learning from Trajectory Demonstrations

As we mention in Section 4.3, HuGE is compatible with learning from trajectory demonstrations. In Figure F.26, we show how HuGE can improve the performance of simple imitation learning starting from different amounts of demonstrations. Given the number of demonstrations, imitation learning fails on less than 10 demonstrations, and with HuGE we can improve the policy to succeed in all cases.

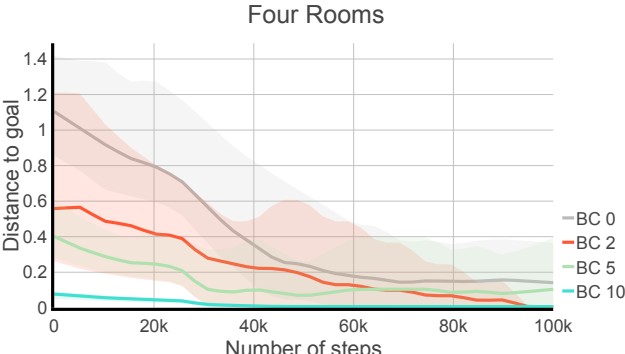

Figure F.26: The figure depicts the distance to the goal in the Four Rooms environment when using a policy pre-trained via Behaviour Cloning with 0, 2, 5, and 10 demonstrations, respectively. We see that using BC on a small number of demonstrations helps to boost the performance of our method. Also, notice that BC wouldn't achieve success (distance $< 0.05$) in any of the cases due to compounding errors which leads to covariant shift. However, HugRL solves these compounding errors within a small number of steps.

# G  Implementation Details

## G.1  Algorithm: stopping before exploration

One important detail of the algorithm is knowing when to stop rolling out the policy and start exploration. The idea is that since the policy is learned in a self-supervised manner, we do not want to have too many redundant steps in a single state. If this happens then because of an imbalance of redundant actions going over this single state, in the training set, will bias the policy to get stuck and not transition to other areas. For it not to happen, we need to stop rolling out the policy at the right moment and launch the next step in the algorithm, the exploration. We propose three different cases where we should stop the rollout, first, once the breadcrumb state $g_b$ is reached, second, once the time-horizon for the episode is reached, third, once the policy is not making any progress and gets stuck in some place. Detecting this can be trivially done looking at the euclidean distance between last states when the state space is in point space, however, it can also be done in image space and we explain it further in Appendix A.1.

## G.2  Hardware

For training the models and running the experiments, we had access to several workstations with one GeForce RTX 2080 Ti or one GeForce RTX 3090. It took on average 8 hours on these machines to run 4 seeds for each one of the baselines and our method. We account the total amount of compute hours would be around 1440 hours for the whole project, taking into account, experimentation and testing the algorithms.

## G.3  Networks with Fourier Features

Seeing the complexity of our benchmarks, where we can have non-smooth reward landscapes for the goal selector. For example, in the four rooms environment, between one side and the other of the right rooms, the reward changes significantly and abruptly. Adding Fourier Features has been shown to work well for fitting these landscapes [56]. For this reason, we used them in some of our experiments, as detailed in Section G. More precisely, when used, we added an additional layer with Fourier features of size $40$ times the input dimension.

## G.4  Training details

The details of the parameters with which the results have been obtained will be disclosed in this section. In particular, Table G.4 depicts the parameters used for the different benchmarks, while Table G.3 contains the hyperparameter configuration used for the different algorithms.

| Parameter | Value |
|---|---|
| *Shared (to those that apply)* | |
| Optimize | Adam |
| Discount factor ($\gamma$) | 0.99 |
| Reward model architecture | MLP$(256, 256)$ |
| Use Fourier in the reward model | True |
| Buffer size reward model | 1000 |
| Steps per reward model update | 1000 |
| *GCSL, Oracle and Ours* | |
| Learning rate | $5 \cdot 10^{-4}$ |
| Batch size | 100 |
| Policy architecture | MLP $(400, 600, 600, 300)$ |
| Steps per policy update | 5000 |
| Use Fourier in the policy model | True |
| Buffer size rollout | 1000 |
| Max gradient norm | 5 |
| Last trajectories to be labeled | 1000 |
| *Human preferences* Same parameters as [52] plus/except | |
| Learning rate | $5 \cdot 10^{-4}$ |
| Batch size | 100 |
| Policy architecture | MLP $(256, 64)$ |
| Steps per policy update | 5000 |
| Use Fourier in the policy model | False |
| Buffer size rollout | 1000 |
| Max gradient norm | 5 |
| Last trajectories to be labeled | 1000 |
| *DDL* | |
| Learning rate | $5 \cdot 10^{-4}$ |
| Batch size | 256 |
| Buffer Size | $2 \cdot 10^{4}$ |
| Policy architecture | MLP $(256, 256)$ |
| Steps per update | 1000 |
| *PPO* Same parameters as [52] plus | |
| Buffer size | 8192 |
| Policy architecture | MLP $(400, 600, 600, 300)$ |

Table G.3: Hyperparameters setting for the algorithms

| Environment | Four rooms | Maze | Pushing around Obstacles | Kitchen | Block Stacking | Bandu |
|---|---|---|---|---|---|---|
| Steps per trajectory | 50 | 250 | 100 | 100 | 10 | 12 |
| Label from last k steps | 10 | 50 | 10 | 20 | 10 | 12 |

Table G.4: Benchmark-related parameters

