# OpenReview forum: "Breadcrumbs to the Goal: Goal-Conditioned Exploration from Human-in-the-Loop Feedback"
_NeurIPS.cc/2023/Conference — NeurIPS 2023 poster_

### Official Review · Reviewer_4hMa · 2023-06-14

**Soundness:** 3 good
**Presentation:** 3 good
**Contribution:** 3 good
**Rating:** 5
**Confidence:** 4

**Summary:**

The paper introduces Human Guided Exploration (HUGE), a system designed to integrate human feedback within the Goal-Conditioned Reinforcement Learning (GCRL) environment, offering a cost-effective and straightforward method for learning diverse robotic tasks through actual human interaction. The authors' primary objective is to strike a balance between overexploration and underexploration in GCRL tasks by incorporating human-in-the-loop assistance. Consequently, human feedback is utilized to ascertain the states requiring further exploration, building on the traditional GCRL algorithms and the GO Explore exploration framework. In particular, a target distance estimation function is trained using binary feedback, facilitating the generation of a more precise subgoal. Subsequently, the Goal-Conditioned Supervised Learning (GCSL) training paradigm is implemented to establish an interaction strategy with the environment. Experimental results in Mujoco and Pybullet demonstrate that HUGE outperforms the preceding GCRL algorithm and Human-in-the-loop algorithm.

**Strengths:**

- The integration of human feedback into the GCRL setting is well-conceived. Hence, the motivation is clear, and the proposed method is, in general, logical and sensible.
- This technique facilitates a simple interface between the human labeler and the algorithm, in which the human supervisor provides binary evaluations to establish which state-goal pairings are closer in comparison to others.

**Weaknesses:**

- The paper presents a degree of innovation that is somewhat restrained, as it essentially combines modified versions of existing methodologies (GCRL, Go Explore, and Human Preference). The foundational GCRL algorithm used is the previously established GCSL, while the exploration component is derived from the Go-Explore paradigm. It is important to note that GCSL paired with Go-Explore could have operated independently without the need for human feedback.
- As per the previous discussion, it's not unexpected to see improved results when human feedback is integrated into the Go-Explore process, as it introduces more a priori knowledge. The implementation of binary feedback can help to decrease the exploration space by learning to rank and establishing a subgoal model.
- This strategy essentially transforms the behavior cloning challenge into a binary human feedback problem. While this appears logical, it makes the motivation somewhat ambiguous and makes specific presumptions about the task at hand. Consequently, the work lacks a degree of originality and novelty. The three fundamental components of the proposed method: the "goal-conditioned policy", "hindsight relabeling", and "learning reward models" from human preference, have all been previously suggested in earlier papers.
-  I think the tasks for evaluation are borderline novel and perhaps not hard enough when we consider a setting where human feedback can be utilized (these tasks are still challenging for RL-based methods learning with dense/sparse rewards). The maze tasks, robotic manipulation tasks are commonly seen in literature (in fact I am surprised PPO is completely failing on some of these tasks, I have personally trained PickNPlace tasks with a RL-based method with a properly shaped reward - Otherwise the authors can also explore other SOTs such as TD3 or SAC) . These tasks are not seen as a fundamental challenge for present learning methods. The main result of the paper is comparison of the proposed method against RL algorithms like PPO and LEXA in terms of sample efficiency. I prefer to see longer horizon tasks where an RL agent with a dense reward cannot easily solve.  I also did not see comparison results for the Sim2Real tasks nor for PickNPlace.

**Questions:**

Other comments:
- The spelling sparce to sparse in Figure 5.

**Limitations:**

-  The work lacks novelty.

---

> ### Author Rebuttal · Authors · 2023-08-10
>
> Thank you for your feedback and for taking the time to review our work. Please find answers/additional experiments to address your concerns below
>
> > “GCSL paired with Go-Explore could have operated independently without human feedback”
>
> Go-Explore + GCSL suffers from the issues of undirected frontier expansion. To illustrate this, we run this baseline (Go-Explore + GCSL) across all domains in Fig AM1. We observe:
>
> - The baseline (purple line, GoExplore+GCSL) does not always succeed and takes more samples than  HuGE.
> - Longer horizon goals are not discovered. In multistep tasks such as the kitchen, Go-Explore + GCSL only manages to discover how to do 1 task (interacting with one of the objects) at a time, it never manages to combine 2 or 3 tasks as HuGE does.
> In contrast, we can see the benefit of HUGE in guiding exploration and reducing search space.
>
> > “It is not unexpected to see improved results when human feedback is integrated into the Go-Explore process”
>
> It is *very important* how we integrate human feedback into the learning process. As we explain in Section E.1 [paragraph: Huge is robust to noisy labels] and show in Fig AM 7 prior work that introduces human feedback into the learning pipeline fails when this human feedback is noisy. We show that using this learned reward function to sample goals and guide exploration makes this method more robust to human feedback.
>
> > “The three fundamental components of the proposed method: the "goal-conditioned policy", "hindsight relabeling", and "learning reward models" from human preference, have all been previously suggested in earlier papers”
>
> While the terms - goal-conditioned policies, hindsight relabeling and learned reward models have been seen in prior papers, we would like to emphasize that:
>
> 1.  We propose a unique way of integrating learned reward models into the exploration of learning goal-conditioned policies.
> 2. *Separating* policy learning from human feedback, and using it for soft frontier expansion (Ours) is crucial to both enable the solving of harder tasks and learning from noisy human feedback. Our results on Go Explore + GCSL [21,24,37], Human Preferences [11], and GCSL [24] emphasize this.
> 3. We enabled capabilities that were not possible in prior work solving long horizon tasks *without* careful reward engineering, using occasional and noisy (crowdsourced) human feedback
>
> > “I think the tasks for evaluation are borderline novel and perhaps not hard enough when we consider a setting where human feedback can be utilized”
>
> We propose and solve harder tasks than any of the prior related work:
> 1. PEBBLE benchmarks: opening single drawers/doors -> HuGE benchmark (kitchen): three sequential tasks in the kitchen involving open drawers, and sliders
> 2. LEXA benchmarks:
>       - two sequential tasks in the kitchen-> HuGE benchmark (kitchen): three sequential tasks in the kitchen involving open drawers, and sliders
>       - Pick and place tasks with two blocks ->Huge benchmark (bandu, block-stacking): assembling a castle-like structure with 4 blocks
>
> HuGE beats both PEBBLE and LEXA (Go-Explore+GCSL) on these harder benchmarks. Moreover, we see that algorithms like PPO/SAC do not learn or learn significantly slower.
>
> > “Surprise that PPO is completely failing on some of these tasks”
>
> The goal in Figure 5 was to provide a fair comparison between all of the benchmarks using the same underlying reward. Since HuGE is robust to noisy underlying reward functions, this works with unshaped reward functions. For further analysis, we performed a new experiment with a more carefully tuned underlying reward function, Fig AM5. We see that PPO and HP fail with totally reasonable reward functions. After engineering effort finetuning these reward functions, these methods succeed, but still slower than HuGE. On the other side, we show HuGE’s robustness to the underlying reward function since it works in both cases.
>
> > “These tasks are not seen as a fundamental challenge for present learning methods” “I prefer to see longer horizon tasks where an RL agent with a dense reward cannot easily solve”
>
> We refer the reviewer to the drawing task in the real world where the reward function for the robot to draw something is very hard to specify. In this case, we do not have access to an oracle dense reward function and yet HuGE is able to succeed from easy-to-provide human feedback.
>
> In addition to this, we have also run state-of-the-art baselines - PPO/SAC and several baselines for learning from human preferences on the environments in simulation. We find that HUGE is both more efficient and performant in these baselines across tasks. This suggests that without significant reward engineering, these tasks *are* challenging for current methods.
>
> > “I also did not see comparison results for the Sim2Real tasks nor for PickNPlace.”
>
> Because of HuGE’s sample efficiency, we can learn the policies directly in the real world, hence, no modeling nor sim2real is required. While this could certainly be performed, the point of the real-world experiments is to show that HUGE is both feedback and sample efficient for real-world learning.
>
> > “This strategy essentially transforms the behavior cloning challenge into a binary human feedback problem.”
>
> The problem we consider is learning from human comparative feedback, not converting behavior cloning into a feedback problem. We propose an algorithm for learning from comparative feedback, and then we show that this algorithm can benefit from some pretraining using behavior cloning. We would appreciate a clarification if we have misunderstood.
>
> > SAC comparison
>
> We also conduct an experiment to compare with other off-policy RL methods like SAC. We see that HuGE and the previous baselines outperform SAC from dense rewards in these benchmarks, Fig AM 1. A small caveat is that these results could be improved somewhat with further tuning of SAC, but will still require reward engineering.

---

> > ### Comment · Reviewer_4hMa · 2023-08-20
> > **Response to the Authors**
> >
> > I would like to thank the authors for their response and efforts towards improving the paper. In the light of reading other reviewers opinion and the additional results, I'm inceasing my score to 5.

---

### Official Review · Reviewer_erJy · 2023-07-03

**Soundness:** 3 good
**Presentation:** 3 good
**Contribution:** 3 good
**Rating:** 7
**Confidence:** 4

**Summary:**

This paper focuses on the exploration problem in decision-making tasks. Previous works try to leverage human guidance with constant synchronous high-quality human feedback, which is expensive and impractical to obtain. In this paper, the authors propose Human Guided Exploration (HUGE), which is able to leverage low-quality feedback from non-expert users. Specifically, the key idea is to separate the challenges of directed exploration and policy learning. Human feedback is only used to direct exploration, while self-supervised policy learning is used to independently learn unbiased behaviors from the collected data. The task for human annotators is just to select which of two states is closer to a particular goal. Then this ranking will be used to train a distance function for goal selection. The experimental results on robotic navigation and manipulation tasks in both simulation and real-world robots demonstrate the advantages of the proposed method.

**Strengths:**

1.	The idea of the proposed method is simple yet effective. Separating the exploration and policy learning dramatically reduces the annotation effort and influence of noisy annotation on policy learning. Using human preference to learn the distance function is also better than directly learning the reward function.
2.	Thorough evaluation of multiple environments, including both simulation and real-world tasks in robotic navigation and manipulation domains. The experimental results show that HUGE outperforms other methods by a large margin.


**Weaknesses:**

1.	Figure 1 provides limited information. It is hard to find that human feedback is noisy and asynchronous. The tasks of pick-and-place and drawing do not contain detailed explanations. This figure can be improved by adding a comparison between previous works that require high-quality data and their method.
2.	The method may need to be evaluated on more complex real-world tasks. The authors argue that Novelty-based exploration performs task-agnostic exploration of the entire state-space, thereby over-exploring the environment. But it seems that the environments used in this paper do not contain very large exploration space.


**Questions:**

No particular questions.

**Limitations:**

No limitation is discussed in the paper. One potential limitation is that the human annotation of the preference of states may be difficult for tasks that require logic reasoning. The experiment environments used in this paper mainly focus on L2 distance, which is easy for humans to quickly select the state close to the goal.

---

> ### Author Rebuttal · Authors · 2023-08-10
>
> Thank you for your feedback and for taking the time to review our work. Please find answers/additional experiments to address your concerns below
>
> > Figure 1 provides limited information.
>
> We will improve this as suggested to include comparisons with prior methods. Thank you for your suggestions!
>
> > The authors argue that Novelty-based exploration performs task-agnostic exploration of the entire state-space, thereby over-exploring the environment. But it seems that the environments used in this paper do not contain very large exploration space.
>
> To verify this, we scaled up the experiment of Go-Explore + Novelty to test task-agnostic exploration via undirected frontier expansion (in Fig AM1). We find that while this can solve certain tasks (maze, four rooms), it takes significantly longer and is unable to solve the more challenging tasks (kitchen, block-stacking, pusher). This shows the presented environments have a large exploration space and also shows why directing frontier expansion is important. Finally, prior work on exploration methods such as LEXA also test their algorithms on some of the same benchmarks, such as the kitchen and block stacking, but they solve shorter horizon problems :
>
> 1. In the kitchen, LEXA only manipulates 2 objects -> we manipulate 3 sequentially,
> 2. In block manipulation LEXA only moves 2 blocks -> we assemble a castle-like structure with 4 blocks
>
> > The method may need to be evaluated on more complex real-world tasks
>
> While this was challenging to perform in the short rebuttal period, we will evaluate this further and we are excited to continue as future work the evaluation on more real-world experiments such as object rearrangement in a household setting, drawing a variety of shapes, etc.
>
> > “One potential limitation is that the human annotation of the preference of states may be difficult for tasks that require logical reasoning. The experiment environments used in this paper mainly focus on L2 distance, which is easy for humans to quickly select the state close to the goal.”
>
> While it is true that many of the environments use distance as an oracle, this is not fundamental to the method, we would like to point out the drawing experiment that we performed in the real world. It is very hard to specify a distance notion for this experiment, however, it remains very easy for the human to quickly select which state is closer to the goal. We will note this potential limitation in the manuscript.

---

> > ### Comment · Reviewer_erJy · 2023-08-11
> > **Response to author**
> >
> > Thanks for addressing my concerns. I don't have further questions.

---

### Official Review · Reviewer_EF3C · 2023-07-06

**Soundness:** 3 good
**Presentation:** 3 good
**Contribution:** 3 good
**Rating:** 7
**Confidence:** 3

**Summary:**

This paper introduces HUGE, a Reinforcement Learning algorithm that makes use of human preferences to guide the selection of partial goals.

HUGE expands upon Goal-Conditional Supervised Learning (GCSL) by improving the goal selection method using noisy labels from humans to form a model, $f_\theta$, of distances to a goal state. This model $f_\theta$ is then used set goals for the GCSL algorithm to learn in a self-supervised fashion, thus bootstrapping learning.

The paper features an extensive evaluation of HUGE, with 6 synthetic tasks, 2 real robot tasks and one task trained from crowd-sourced feedback. Results show that HUGE outperforms a large number of baselines.

**Strengths:**

* HUGE combines intuitions from GCSL and GoExplore, and can take advantage of asynchronous human preferences.
* The paper is really well written and is easy to follow.
* The paper presents a thorough evaluation. I find it particularly that HUGE Was able to train a real robot policy in under 30 hours with just 130 annotations.
* HUGE can take advantage, but does not require, expert demonstrations through Behaviour Cloning.

**Weaknesses:**

* It would have been more representative to choose PEBBLE [1] rather than Christiano et al. (2017) as a more representative example of the performance of Preference-based Reinforcement Learning.
* Some of the results are buried in the appendix. For instance section 5.2 does not mention what was the task analysed are. From Figure 6. left, I assume it's the Kitchen task from D4RL. Still, neither the final performance nor the task are mentioned in the main text.
    * Similarly, the main results of the analysis comparing the final performance and the number of annotations should be included in the main text, since it provides readers with an estimation of how many labels they may need for their task.

**Post-rebuttal update**

Authors added a comparison against PEBBLE, where HUGE outperforms it.

**Questions:**

1. How were the annotations for the real-world tasks obtained?
2. Why does BC+Ours finish early in Figure 5 Kitchen & Bandu?

**Minor nitpicks and suggestions:**
* $\mathcal{G}$ is undefined in equation 4.
* In algorithm 1, step 5, the second argument to PolicyExploration should be $f_\theta$.
* In figure 6 left, what is the difference between "human + 5 demos" and "crowd-source + 5 demos". Moreover, the colours of their curves are indistinguishable to my eyes.
* When defining $f_\theta$, I would clarify that "closer" is not necessarily meant as a mathematical distance, but rather as a human intuition of closeness, which I belive is what $f_\theta$ is supposed to capture.


**Post-rebuttal update**

See discussion for answers to the above questions.

**Limitations:**

The limitations have been adequately addressed.

---

> ### Author Rebuttal · Authors · 2023-08-10
>
>
> Thank you for your feedback and for taking the time to review our work. Please find the answers to your comments and concerns below.
>
> > “More representative to choose PEBBLE”
>
> We implemented PEBBLE (using author provided GitHub repository) and provide the new curves with this implementation in Fig AM1. We find that while PEBBLE can learn in certain environments (block stacking, bandu, maze), however, the performance is worse than Human Preferences and HuGE.
>
> Since the reviewer seemed particularly keen on comparing to PEBBLE we tried to understand better why it was failing and we find the following:
>
> - We observe that when the initial unsupervised exploration step does not cover a big enough set of the state space, PEBBLE gets stuck and does not manage to reach the goal.
>
> - We spent a decent amount of time trying to make it work better (hyperparameter tuning on entropy, number of unsupervised steps, increasing the stochasticity in the rollout actions) however nothing seemed to make it work on our benchmarks. We used the implementation provided by the authors and we also implemented our own version, in both cases this faced the same issues of getting stuck mid-way (meaning exploration collapsed and no progress was made).
>
> > “Some of the results are buried in the appendix”
>
> As suggested, we will move the final performance of the baselines, and the number of annotations for the real human experiments into the main text of the paper and provide more clarity on the problem setting.
>
> > “For instance, section 5.2 does not mention what the task analyzed are.”
>
> Thank you for pointing this out! It was only mentioned in small in Figure 6’s title and we will clarify this in the camera-ready version. In the meantime for more information, the crowdsourced data collection was performed on the kitchen manipulation task, where the franka arm needs to open 3 objects (cabinet, microwave, slider), more details about the kitchen environment on Appendix C.
>
> > “The main results of the analysis comparing the final performance and the number of annotations should be included in the main text”
>
> Indeed. We will include this in the main text in Section 5.1. In the meantime, we refer the reviewers to Appendix D and Table D.10 for more details.
>
> > “How were the annotations for the real-world tasks obtained?”
>
> We obtained the annotations for the real-world task in the same manner as in the simulated experiments, querying three annotators, no crowdsourcing was done in these experiments. We will clarify this in the text, and for more information on how the feedback was collected on the simulated experiments we refer the reviewers to Appendix B.
>
> > “Why does BC+Ours finish early in Figure 5 Kitchen & Bandu?”
>
> We will update these two curves for the camera-ready, they were not fully completed due to computing limitations.
>
> >“In figure 6 left, what is the difference between "human + 5 demos" and "crowd-source + 5 demos". ”
>
> The difference is in the number of humans that collected the labels, for “human + 5 demos” only one annotator provided the labels, for “crowd-source + 5 demos” 109 annotators provided the feedback, we provide more details about this crowdsourcing experiment in Appendix B.1.
>
> > Minor corrections
>
> We thank you for this feedback, and we will incorporate this into the manuscript.

---

> > ### Comment · Reviewer_EF3C · 2023-08-15
> > **Thank you for the rebuttal.**
> >
> > Thank you for your detailed rebuttal, the additional experiments regarding PEBBLE and for really trying to get the PEBBLE baseline working. It makes sense that insufficient unsupervised exploration would be cause PEBBLE to collapse.
> >
> > Since my main concern and questions have all been addressed, I will increase my rating.

---

### Official Review · Reviewer_MJKZ · 2023-07-07

**Soundness:** 4 excellent
**Presentation:** 2 fair
**Contribution:** 3 good
**Rating:** 6
**Confidence:** 4

**Summary:**

Broadly, the paper addresses the challenge of learning multi-stage robotic navigation and manipulation tasks in simulation.

The paper frames several benchmark tasks as goal-conditioned RL (GCRL), and they present a novel method ("HUGE") for leveraging human preferences collected during learning. In particular, the preferences are used to train a goal-selector model that guides (biases) exploration. For policy-learning, they use goal-conditioned supervised learning on the collected replay buffer, without requiring a handcrafted task-specific reward.

Compared to [1], HUGE also uses hindsight relabeling to learn from a replay buffer despite a sparse reward, but it differs in that it biases sampling from the replay buffer based on human preferences. Compared to [11], HUGE also collects human preferences iteratively (intertwined with learning). However, HUGE doesn't use its collected human feedback to learn a reward or otherwise directly bias the policy. Instead, the feedback is only used to bias exploration. Compared to [17], HUGE also has an exploration phase that builds an archive of trajectories. However, HUGE chooses promising goal states and generates trajectories using its goal-condition policy, while [17] chooses promising start states based on novelty and can generate trajectories with various exploration policies (random, epsilon-greedy, novelty-based).

The authors evaluate on four simulated manipulation tasks, two simulated navigation tasks, and two real-robot tasks. They compare to several baselines including the related work mentioned above as well as PPO (with dense and sparse reward variants).

**Strengths:**

Their method doesn't require a handcrafted reward and it's able to leverage noisy, infrequent human feedback. These strengths are relevant and valuable to this domain.

Evaluation on a variety of simulated and real tasks.

Impressive crowdsourcing infrastructure and diversity of annotators.

Interesting, well-explained adaption of real-world tasks to fit their use of goal-conditioned RL.

**Weaknesses:**

Lack of clear, fair numeric results on task performance across HUGE and the baselines:

= In Figure 5, I can only compare task performance after a given number of steps. A table with a performance number after training each approach to convergence would be more clear and would better support the claim from your text "HUGE beats prior work on the proposed long horizon tasks in simulation".

= It's not clear why some tasks were trained for millions of steps and others not.

= The paper text conveys that Figure 5's various approaches are using a poorly-tuned reward function ("we did not do any reward finetuning"). In particular, this makes the PPO results questionable.

= This section also uses synthetic human preferences derived from your reward function, instead of real human preferences. This confuses me and contradicts the earlier premise of HUGE. HUGE with this synthetic source of preferences would appear to be another RL method that utilizes a handcrafted reward function and not human feedback. This muddies your comparison with other approaches, especially [11] which is intended to be guided by real (not synthetic) human feedback.

= It seems like other baselines might benefit from the addition of behavior-cloning pretraining, as you did for "BC + Ours".

**Questions:**

Can you improve your comparison to baselines based on the feedback I gave above? In particular, I am most interested in a robust, fair comparison to [11] ("Human Preferences"), because I consider that work the most directly comparable to yours: it doesn't require a handcrafted reward signal but it does require collection of human preferences intertwined with learning.

I hope you'll address the weaknesses in your PPO baseline I mentioned earlier: train to convergence with a fine-tuned reward function. Figure 5 currently shows that HUGE (with synthetic preferences derived from a reward function) outperforms PPO using the same reward function. If this result holds after you improve the PPO baseline, can you analyze this? Would it mean the synthetic variant of HUGE is fundamentally a better optimization algorithm compared to PPO?

I notice your two real-world tasks are fully-observable. How would you apply your approach to a real-world task that is only partially-observable, such as room-scale manipulation with visual perception?

**Limitations:**

No concerns here

---

> ### Author Rebuttal · Authors · 2023-08-10
>
> Thank you for your feedback and for taking the time to review our work. Please find answers/additional experiments to address your concerns below
>
> > “Poorly-tuned reward function”
>
> A key point of our paper is the robustness to noisy human feedback, which means that even with a simple guiding human proxy, HuGE can still solve the proposed tasks, as we showed in Appendix E.15. It is known that this is not the case with PPO which needs careful reward shaping to succeed AM[3,4], and also reflected in our results in Fig 5 in the original paper.
>
> We provide a new analysis experiment on this result (Fig AM 5), where we test HuGE, PPO, and Learning from Human Preferences (using PPO) on two reward functions, one that we very carefully shaped during the rebuttal period, and another that is as in the original submission (note that the reward function serves to inform the synthetic human proxy for HuGE). We observe:
>
> - HuGE succeeds with both reward functions
>
> - PPO only succeeds when the reward function is carefully shaped.
>
> - Human Preferences only partially succeeds when the reward function is carefully shaped
>
> - When PPO and Human Preferences succeed, they are significantly less sample efficient than HUGE.
>
> This demonstrates that HuGE is robust to simple underlying reward functions, which makes it robust to learning these reward functions from Human Feedback. This is also reinforced by additional experiments with real human feedback on all domains as shown in Fig AM 2-3.
>
> >“Going from Synthetic Human Feedback to Real Human Feedback” & “Fair comparison with Human Preferences”
>
> In Figure 5, all baselines that rely on human feedback were trained using a proxy-human for a fair comparison among methods. However, as the reviewer requested in Figure AM 2-3 we provide additional results of HuGE working from real human feedback and results on Human Preferences from real human feedback on the four rooms and empty rooms environments. With this robust, fair comparison between HuGE and Human Preferences, we observe that when learning from real humans:
>
> - HuGE is more sample efficient in the number of human labels compared to learning from   human preferences [11] with real humans
>
> - HuGE is more sample efficient in the number of timesteps to succeed compared to [11].
>
> >“Analysis of PPO vs HuGE (from synthetic feedback)”
>
> As noted, we observe that HuGE is more sample efficient than PPO. To analyze why this is, we ran a comparison on gradient variance using HUGE as compared to gradient variance with PPO, comparing the average pairwise cosine similarity across mini-batches to measure gradient variance (as suggested in AM[1]). We find that training the gradient similarity is significantly larger in HuGE than in PPO, potentially explaining improved sample complexity. Besides this, other components in HuGE may also lead to faster learning:
>
> 1. Generalization across states that come from self-supervised hindsight relabeling,
> 2. Hindsight relabeled policy learning is off-policy while PPO is on-policy.
>
> > “Does it mean the synthetic variant of HuGE is fundamentally a better optimization algorithm compared to PPO?”
>
> Not necessarily, given a carefully shaped reward function, PPO typically eventually works on most decision-making problems, however, HuGE is a method for goal-reaching problems, which is a subset of all possible decision-making problems. In these goal-reaching problems, we have shown that HuGE can typically outperform PPO even at convergence when the reward functions are noisy. However, PPO is a more general learning paradigm and more careful experimentation and ablations are needed before making broad-reaching claims.
>
> >“Other Baselines would benefit from BC pretraining”
>
> To provide a fair comparison, we also added learning from human preferences with the same amount of BC pretraining (implemented using stable baselines AM[2]). We found that despite this pretraining, learning from human preferences [11] and PPO were outperformed by HUGE.
>
> >“Table of performance at convergence:”
>
>  We would like to refer the reviewer to Table D.10 with the results after convergence in the Appendix.
>
> >“Number of training steps per task”
>
>  Each task has different action and observation spaces as well as different ranges of timesteps per episode, we refer the reviewer to Appendix C for more details. Also, we ran each baseline until convergence.
>
> >“How would you apply your approach to a real-world task that is only partially observable”
>
> As of now, the framework is largely structured around fully observable problems, as is a significant body of work on reinforcement learning in MDPs [11,21,24,37]. Training recurrent policies, rather than MLPs, as well as recurrent goal selectors, may make the system applicable to partially observable settings.
>
> AM[1] Ilyas, Andrew, et al. "A closer look at deep policy gradients." arXiv preprint arXiv:1811.02553 (2018)
>
> AM[2] Raffin, Antonin, et al. "Stable-Baselines3: Reliable Reinforcement Learning Implementation" http://jmlr.org/papers/v22/20-1364.html (2021)
>
> AM[3] Learning robust perceptive locomotion for quadrupedal robots in the wild, Takahiro Mki, et al, Science Robotics, 2022.
>
> AM[4] Visual Dexterity: In-hand Dexterous Manipulation from Depth, Tao Chen, et al. 2022, arxiv.org/abs/2211.11744

---

> > ### Comment · Reviewer_MJKZ · 2023-08-16
> >
> > Thanks for the detailed rebuttal. The main novel contribution here is the use of a noisy preference signal to bias exploration, while avoiding biasing the learned policy. The rebuttal experiments and baselines show more clearly that this works well, regardless of whether this is synthetic (e.g. a poorly-tuned reward function) or real human preferences. I've revised my rating.

---

### Author Rebuttal · Authors · 2023-08-10

Dear reviewers,

Thank you for your constructive feedback. In response to reviewer concerns, we have conducted a number of new experiments and analyses. We describe these briefly below and refer reviewers to individual responses for detailed clarifications:

**[DISCLAIMER 1]**: In the following experiments we talk about reward functions and RL methods that use reward functions. However, HuGE does *NOT* need any reward function, it learns uniquely from human feedback. Where specified, we use human proxies, which have an underlying reward function, to provide a fair comparison among different baselines.

**[DISCLAIMER 2]**: Figures in the Aditional Material (PDF) attached to the rebuttal are referred in the text as Figure AM ${figure number}.

1. **More carefully tuned rewards for PPO**: (reviewer MJKZ): We compare the performance of HuGE, PPO, and Human Preferences from finetuned/not finetuned reward functions in order to clarify how important careful reward engineering is to the success of these methods. Findings:
    - HUGE succeeds even without careful reward engineering,
    - PPO and Human preferences struggle without careful reward engineering.
    - With careful reward engineering - PPO and Human preferences can succeed, but they learn significantly slower than HUGE. (Fig AM 5)

2. **HUGE (Ours) with real human feedback (reviewer MJKZ)**: We provide additional results on running HuGE from real human feedback across more benchmarks (see Fig AM 2-3). HuGE matches performance when feedback is provided by a synthetic human.

3. **Human preferences [11] with real human feedback (reviewer MJKZ)**: To induce a fair comparison, we also run learning from human preferences[11] with real human feedback, as shown in Fig AM2. Human preferences:
     - Needs more human labels to converge
     - More timesteps to converge
     - converges to lower performance compared to HuGE

4. **HuGE is more robust to noisy feedback compared to Human Preferences** (reviewer 4hMa)
     - HuGE matches the behavior obtained with perfect annotations whilst Human Preferences does not solve the task at hand, see Fig AM 7

5. **BC pre-training with human preferences** (reviewer MJKZ): To induce a fair comparison when pretraining data is available, we introduced BC pretraining on a couple of benchmarks for PPO and Human Preferences (see Fig AM 1) and compared this to HUGE with pretraining. We found that even if each method improves when adding demo pretraining, these still take longer than HuGE with and without demos to converge.

6. **Analysis of PPO vs HuGE** (reviewer MJKZ): Since HUGE is much more sample efficient than PPO, we conducted analysis to understand why this might be the case for goal reaching problems. We hypothesize this may be (atleast in part) due to lower variance gradients, and show this in Fig AM 6.

7. **PEBBLE comparison** (reviewer EF3C): We conducted an experiment to compare with an off-policy RL method learning from human preferences such as PEBBLE [31]. We find:
      - HuGE and Human Preferences methods outperform PEBBLE

8. **Go-Explore + GCSL** (reviewer 4hMa): We conducted an experiment to understand whether GCSL + go-explore by itself is effective enough to solve the proposed tasks. We find in Fig AM1 that Go-Explore + GCSL:
      - Longer horizon goals are not discovered (such as in the kitchen benchmark), which is also seen in prior work (LEXA)[34].
      - When it solves the task it takes much longer than HuGE to discover target goals since exploration is undirected.
      - SAC comparison (reviewer 4hMa): We conducted an experiment to compare with other off-policy RL methods like SAC. We see that HuGE and the previous baselines outperform SAC from dense rewards in these benchmarks, Fig AM 1.

Please let us know if other experiments/clarifications can help with the discussion!

---

### Decision · Program_Chairs · 2023-09-21

**Decision:**

Accept (poster)

**Comment:**

The submission here describes an interesting, novel, and practical method for leveraging human feedback to help with exploration in reinforcement learning. The presentation of the work is excellent and the empirical evaluation and results are impressive. The work is of sufficient quality for presentation at NeurIPS. The authors should now spend the time to succinctly update the submission to include the extra results and discussion points that proved so effective during the author-reviewer discussion period.